# A Model or 603 Exemplars: Towards Memory-Efficient Class-Incremental Learning

**Da-Wei Zhou, Qi-Wei Wang, Han-Jia Ye,** De-Chuan Zhan
State Key Laboratory for Novel Software Technology, Nanjing University
`{zhoudw, wangqiwei, yehj, zhandc}@lamda.nju.edu.cn`

## Abstract

Real-world applications require the classification model to adapt to new classes without forgetting old ones. Correspondingly, Class-Incremental Learning (CIL) aims to train a model with *limited memory size* to meet this requirement. Typical CIL methods tend to save representative exemplars from former classes to resist forgetting, while recent works find that storing models from history can substantially boost the performance. However, the stored models are not counted into the memory budget, which implicitly results in unfair comparisons. We find that when counting the model size into the total budget and comparing methods with aligned memory size, saving models do not consistently work, especially for the case with limited memory budgets. As a result, we need to holistically evaluate different CIL methods at different memory scales and simultaneously consider accuracy and memory size for measurement. On the other hand, we dive deeply into the construction of the memory buffer for memory efficiency. By analyzing the effect of different layers in the network, we find that shallow and deep layers have different characteristics in CIL. Motivated by this, we propose a *simple yet effective baseline*, denoted as Memo for Memory-efficient Expandable MOdel. Memo extends specialized layers based on the shared generalized representations, efficiently extracting diverse representations with modest cost and maintaining representative exemplars. Extensive experiments on benchmark datasets validate Memo's competitive performance. Code is available at: `https://github.com/wangkiw/ICLR23-MEMO`

## 1 Introduction

In the open world, training data is often collected in stream format with new classes appearing (Gomes et al., 2017; Geng et al., 2020). Due to storage constraints (Krempl et al., 2014; Gaber, 2012) or privacy issues (Chamikara et al., 2018; Ning et al., 2021), a practical Class-Incremental Learning (CIL) (Rebuffi et al., 2017) model requires the ability to update with incoming instances from new classes without revisiting former data. The absence of previous training data results in *catastrophic forgetting* (French, 1999) in CIL — fitting the pattern of new classes will erase that of old ones and result in a performance decline. The research about CIL has attracted much interest (Zhou et al., 2021b; 2022a;b; Liu et al., 2021b; 2023; Zhao et al., 2021a;b) in the machine learning field.

Saving all the streaming data for offline training is known as the performance *upper bound* of CIL algorithms, while it requires an unlimited memory budget for storage. Hence, in the early years, CIL algorithms are designed in a *strict* setting without retaining any instances from the former classes (Li & Hoiem, 2017; Kirkpatrick et al., 2017; Aljundi et al., 2017; Lee et al., 2017). It only keeps a classification model in the memory, which helps save the memory budget and meanwhile preserves privacy in the deployment. Afterward, some works noticed that saving *limited exemplars* from former classes can boost the performance of CIL models (Rebuffi et al., 2017; Chaudhry et al., 2019). Various exemplar-based methodologies have been proposed, aiming to prevent forgetting by revisiting the old during new class learning, which improves the performance of CIL tasks steadily (Rolnick et al., 2019; Castro et al., 2018; Wu et al., 2019; Isele & Cosgun, 2018). The utilization of exemplars has drawn the attention of the community from the strict setting to update the model with *restricted* memory

---

*Correspondence to: Han-Jia Ye (yehj@lamda.nju.edu.cn)

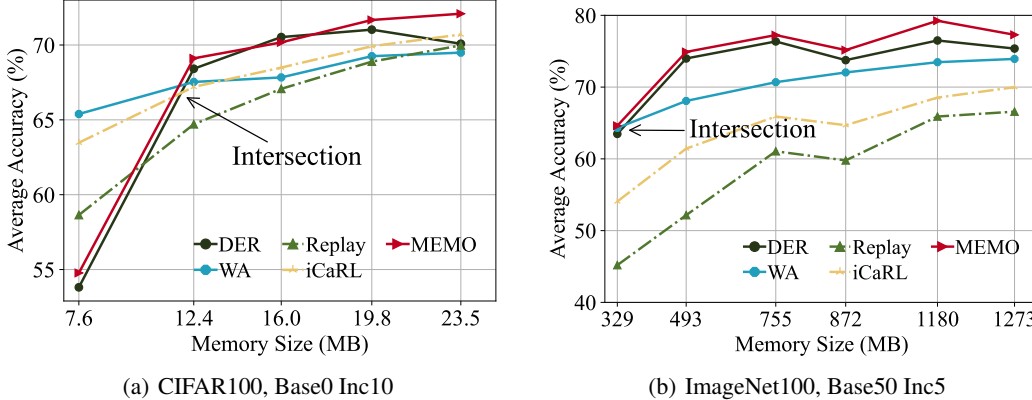

(a) CIFAR100, Base0 Inc10          (b) ImageNet100, Base50 Inc5

Figure 1: The average accuracy of different methods by varying memory size from small to large. The start point corresponds to the memory size of exemplar-based methods with benchmark backbone (WA (Zhao et al., 2020), iCaRL (Rebuffi et al., 2017), Replay (Chaudhry et al., 2019)), and the endpoint corresponds to the memory cost of model-based methods with benchmark backbone (DER (Yan et al., 2021) and MEMO (our proposed method)). We align the memory cost by using the small model for model-based methods or adding exemplars for exemplar-based methods. 'Base' stands for the number of classes in the first task, and 'Inc' represents the number of classes in each incremental new task. See Section 4.1 and 4.2 for more details.

size (Castro et al., 2018; Rebuffi et al., 2017). Rather than storing exemplars, recent works (Yan et al., 2021; Wang et al., 2022; Li et al., 2021; Douillard et al., 2021) find that *saving backbones* from the history pushes the performance by one step towards the upper bound. These *model-based* methods propose to train multiple backbones continually and aggregate their representations as the feature representation for final prediction. Treating the backbones from history as 'unforgettable checkpoints,' this line of work suffers less forgetting with the help of these diverse representations.

Model-based CIL methods push the performance towards the upper bound, but does that mean catastrophic forgetting is solved? Taking a close look at these methods, we find that they implicitly introduce an extra memory budget, namely *model buffer* for keeping old models. The additional buffer implicitly results in an unfair comparison to those methods without storing models. Take CIFAR100 (Krizhevsky et al., 2009) for an example; if we exchange the model buffer of ResNet32 (He et al., 2015) into exemplars of equal size and append them to iCaRL (Rebuffi et al., 2017) (a baseline without retaining models), the average accuracy drastically improves from 62% to 70%. How to fairly measure the performance of these methods remains a long-standing problem since saving exemplars or models will both consume the memory budget. In this paper, we introduce an *extra dimension* to evaluate CIL methods by considering both incremental performance and memory cost. For those methods with different memory costs, we need to *align* the performance measure at the same memory scale for a fair comparison.

**How to fairly compare different methods?** There are two primary sources of memory cost in CIL, *i.e.*, *exemplar and model buffer*. We can align the memory cost by switching the size of extra backbones into extra exemplars for a fair comparison. For example, a ResNet32 model has the same memory size with 603 images for CIFAR100, and 297 ImageNet (Deng et al., 2009) images have the same memory size with a ResNet18 backbone. Figure 1 shows the fair comparison on benchmark datasets, *e.g.*, CIFAR100 and ImageNet100. We report the average accuracy of different models by *varying the memory size from small to large*. The memory size of the start point corresponds to the cost of an exemplar-based method with a single backbone, and the endpoint denotes the cost of a model-based method with multiple backbones. As we can infer from these figures, there is an **intersection** between these methods — saving models is less effective when the total budget is limited while more effective when the total budget is ample.

In this paper, we dive deeply into the empirical evaluations of different CIL methods considering the incremental performance and memory budget. Towards a fair comparison between different approaches, we propose several new measures that simultaneously consider performance and memory size, *e.g.*, area under the performance-memory curve and accuracy per model size. On the other hand, **how to organize the memory buffer efficiently so that we can save more exemplars and meanwhile maintain diverse representations?** We analyze the effect of different layers of the network by counting the gradients and shifting range in incremental learning, and find that shallow layers tend to learn generalized features. By contrast, deep layers fit specialized features for corresponding tasks and yield very different characteristics from task to task. As a result, sharing the shallow layers and only creating deep layers for new tasks helps save the memory budget in CIL.

Furthermore, the spared space can be exchanged for an equal number of exemplars to further boost the performance. Intuitively, we propose a *simple yet effective* baseline MEMO to simultaneously consider extending diverse features with the most modest memory cost. MEMO shows competitive results against state-of-the-art methods under the fair comparison on vast benchmark datasets and various settings, which obtains the best performance in most cases of Figure 1.

## 2   RELATED WORK

We roughly divide current CIL methods into two groups, *i.e.*, exemplar-based and model-based methods. The former group seeks to rehearse former knowledge when learning new, and the latter saves extra model components to assist incremental learning. Obviously, some methods do not fall into these two groups (Kirkpatrick et al., 2017; Li & Hoiem, 2017; Jin et al., 2021; Smith et al., 2021), and we refer the readers to (Zhou et al., 2023) for a holistic review.

**Exemplar-Based Methods:** Exemplars are representative instances from former classes (Welling, 2009), and CIL models can selectively save a relatively small amount of exemplars for rehearsal during updating (Isele & Cosgun, 2018). Like natural cognitive systems, rehearsal helps revisit former tasks to resist catastrophic forgetting (Parisi et al., 2019). Apart from direct replay, there are other methods addressing utilizing exemplars in CIL. iCaRL (Rebuffi et al., 2017) builds knowledge distillation (Zhou et al., 2003; Zhou & Jiang, 2004; Hinton et al., 2015) regularization with exemplars to align the predictions of old and new models. On the other hand, (Lopez-Paz & Ranzato, 2017) treats the loss on the exemplar set as an indicator of former tasks' performance and solves the quadratic program problem as regularization. (Wu et al., 2019; Castro et al., 2018) utilize exemplars for balanced finetuning or bias correction. Note that exemplars can be directly saved or be generated with extra generative models (Shin et al., 2017b; He et al., 2018), which indicates the *equivalency* between models and exemplars. Consistent with our intuition, there has been much research addressing that saving more exemplars will improve the performance of CIL models correspondingly (Iscen et al., 2020; Ahn et al., 2021).

**Model-Based Methods:** There are some methods that consider increasing model components incrementally to meet the requirements of new classes. (Liu et al., 2021a) adds residual blocks as mask layers to balance stability and plasticity. (Ostapenko et al., 2021) expands dynamic modules with dynamic routing to generalize to related tasks. (Yoon et al., 2018) creates new neurons to depict the features for new classes when needed, and (Xu & Zhu, 2018) formulates it as a reinforcement learning problem. Instead of expanding the neurons, some works (Serra et al., 2018; Rajasegaran et al., 2019; Abati et al., 2020) propose to learn masks and optimize the task-specific sub-network. These methods increase a modest amount of parameters to be optimized. Recently, (Yan et al., 2021) addresses that aggregating the features by training a single backbone for each incremental task can substantially improve the performance. Since there could be numerous incremental tasks in CIL, saving a backbone per task *implicitly* shifts the burden of storing exemplars into retaining models.

**Memory-Efficient CIL:** Memory cost is an important factor when deploying models into real-world applications. (Iscen et al., 2020) addresses saving extracted features instead of raw images can help model learning. Similarly, (Zhao et al., 2021c) proposes to save low-fidelity exemplars to reduce the memory cost. (Smith et al., 2021; Choi et al., 2021) release the burden of exemplars by data-free knowledge distillation (Lopes et al., 2017). To our knowledge, we are the first to address the memory-efficient problem in CIL from the model buffer perspective.

## 3   PRELIMINARIES

### 3.1   PROBLEM DEFINITION

Class-incremental learning was proposed to learn a stream of data continually with new classes (Rebuffi et al., 2017). Assume there are a sequence of $B$ training tasks $\left\{\mathcal{D}^1, \mathcal{D}^2, \cdots, \mathcal{D}^B\right\}$ without overlapping classes, where $\mathcal{D}^b = \left\{\left(\mathbf{x}_i^b, y_i^b\right)\right\}_{i=1}^{n_b}$ is the $b$-th incremental step with $n_b$ instances. $\mathbf{x}_i^b \in \mathbb{R}^D$ is a training instance of class $y_i \in Y_b$, $Y_b$ is the label space of task $b$, where $Y_b \cap Y_{b'} = \varnothing$ for $b \neq b'$. A fixed number of representative instances from the former classes are selected as exemplar set $\mathcal{E}$, $|\mathcal{E}| = K$ is the fixed exemplar size. During the training process of task $b$, we can only access data from $\mathcal{D}^b$ and $\mathcal{E}$. The aim of CIL at each step is not only to acquire the knowledge from the current task $\mathcal{D}^b$ but also to preserve the knowledge from former tasks. After each task, the trained

model is evaluated over all seen classes $\mathcal{Y}_b = Y_1 \cup \cdots Y_b$. The incremental model is unaware of the task id, *i.e.*, $b$, during inference. We decompose the model into the embedding module and linear layers, *i.e.*, $f(\mathbf{x}) = W^\top \phi(\mathbf{x})$, where $\phi(\cdot) : \mathbb{R}^D \to \mathbb{R}^d$, $W \in \mathbb{R}^{d \times |\mathcal{Y}_b|}$.

## 3.2 OVERCOME FORGETTING IN CLASS-INCREMENTAL LEARNING

In this section, we separately introduce two baseline methods in CIL. The former baseline belongs to the exemplar-based method, while the latter belongs to the model-based approach.

**Knowledge Distillation:** To make the updated model still capable of classifying the old class instances, a common approach in CIL combines cross-entropy loss and knowledge distillation loss (Zhou et al., 2003; Zhou & Jiang, 2004; Hinton et al., 2015). It builds a mapping between the former and the current model:

$$\mathcal{L}(\mathbf{x}, y) = (1 - \lambda) \underbrace{\sum_{k=1}^{|\mathcal{Y}_b|} -\mathbb{I}(y = k) \log \mathcal{S}_k(W^\top \phi(\mathbf{x}))}_{\textit{Cross Entropy}} + \lambda \underbrace{\sum_{k=1}^{|\mathcal{Y}_{b-1}|} -\mathcal{S}_k(\bar{W}^\top \bar{\phi}(\mathbf{x})) \log \mathcal{S}_k(W^\top \phi(\mathbf{x}))}_{\textit{Knowledge Distillation}}, \quad (1)$$

where $\mathcal{Y}_{b-1} = Y_1 \cup \cdots Y_{b-1}$ denotes the set of old classes, $\lambda$ is trade-off parameter, and $\mathcal{S}_k(\cdot)$ denotes the $k$-th class probability after softmax operation. $\bar{W}$ and $\bar{\phi}$ correspond to frozen classifier and embedding before learning $\mathcal{D}^b$. Aligning the output of the old and current models helps maintain discriminability and resist forgetting. The model optimizes Eq. 1 over the current dataset and exemplar set $\mathcal{D}^b \cup \mathcal{E}$. It depicts a simple way to simultaneously consider learning new class and preserving old class knowledge, which is widely adopted in exemplar-based methods (Rebuffi et al., 2017; Wu et al., 2019; Zhao et al., 2020).

**Feature Aggregation:** Restricted by the representation ability, methods with a single backbone cannot depict the dynamic features of new classes. For example, if the first task contains 'tigers,' the CIL model will pay attention to the features to trace the beards and stripes. If the next task contains 'birds,' the features will then be adjusted for beaks and feathers. Since a single backbone can depict a *limited number of* features, learning and overwriting new features will undoubtedly trigger the forgetting of old ones. To this end, DER (Yan et al., 2021) proposes to add a backbone to depict the new features for new tasks. For example, in the second stage, it initializes a new feature embedding $\phi_{new} : \mathbb{R}^D \to \mathbb{R}^d$ and freezes the old embedding $\phi_{old}$. It also initializes a new linear layer $W_{new} \in \mathbb{R}^{2d \times |\mathcal{Y}_b|}$ inherited from the linear layer $W_{old}$ of the last stage, and optimizes the model with typical cross-entropy loss:

$$\mathcal{L}(\mathbf{x}, y) = \sum_{k=1}^{|\mathcal{Y}_b|} -\mathbb{I}(y = k) \log \mathcal{S}_k(W_{new}^\top [\bar{\phi}_{old}(\mathbf{x}), \phi_{new}(\mathbf{x})]) \,. \quad (2)$$

Similar to Eq. 1, the loss is optimized over $\mathcal{D}^b \cup \mathcal{E}$, aiming to learn new classes and remember old ones. It also includes an auxiliary loss to differentiate the embeddings of old and new backbones, which will be discussed in the supplementary. Eq. 2 sequentially optimizes the newest added backbone $\phi_{new}$ and fixes old ones. It can be seen as fitting the residual term to obtain diverse feature representations among all seen classes. Take the aforementioned scenario for an example. The model will first fit the features for beards and strides to capture tigers in the first task with $\phi_{old}$. Afterward, it optimizes the features for beaks and feathers to recognize birds in the second task with $\phi_{new}$. Training the new features will not harm the performance of old ones, and the model can obtain diverse feature representations as time goes by. However, since it creates a new backbone per new task, it requires saving all the embeddings during inference and consumes a much larger memory cost compared to exemplar-based methods.

## 4 ANALYSIS

### 4.1 EXPERIMENTAL SETUP

As our paper is heavily driven by empirical observations, we first introduce the three main datasets we experiment on, the neural network architectures we use, and the implementation details.

**Dataset**: Following the benchmark setting (Rebuffi et al., 2017; Wu et al., 2019), we evaluate the performance on **CIFAR100** (Krizhevsky et al., 2009), and **ImageNet100/1000** (Deng et al., 2009). CIFAR100 contains 50,000 training and 10,000 testing images, with a total of 100 classes. Each image is represented by $32 \times 32$ pixels. ImageNet is a large-scale dataset with 1,000 classes, with about 1.28 million images for training and 50,000 for validation. We also sample a subset of 100 classes according to (Wu et al., 2019), denoted as ImageNet100.

**Dataset Split:** According to the common setting in CIL (Rebuffi et al., 2017), the class order of training classes is shuffled with random seed 1993. There are two typical class splits in CIL. The former (Rebuffi et al., 2017) equally divides all the classes into $B$ stages. The latter (Hou et al., 2019; Yu et al., 2020) treats half of the total classes in the first stage (denoted as base classes) and equally divides the rest classes into the incremental stages. Without loss of generality, we use **Base-$x$, Inc-$y$** to represent the setting that treats $x$ classes as base classes and learns $y$ new classes per task. $x = 0$ denotes the former setting.

**Implementation Details:** All models are deployed with PyTorch (Paszke et al., 2019) and Py-CIL (Zhou et al., 2021a) on NVIDIA 3090. If not specified otherwise, we use the *same* network backbone (Rebuffi et al., 2017) for *all* compared methods, *i.e.*, ResNet32 (He et al., 2015) for CI-FAR100 and ResNet18 for ImageNet. The model is trained with a batch size of 128 for 170 epochs, and we use SGD with momentum for optimization. The learning rate starts from $0.1$ and decays by $0.1$ at 80 and 150 epochs. The source code of MEMO will be made publicly available upon acceptance. We use the herding (Welling, 2009) algorithm to select exemplars from each class.

## 4.2 HOW TO FAIRLY COMPARE CIL METHODS?

As discussed in Section 3.2, exemplar-based methods and model-based methods consume different memory sizes when learning CIL tasks. Aiming for a fair comparison, we argue that these methods should be *aligned* to the same memory cost when comparing the results. Hence, we vary the total memory cost from small to large and compare different methods at these selected points. Figure 1 shows corresponding results, which indicates the performance *given different memory budget*. Take Figure 1(a) for an example; we first introduce how to set the start point and endpoint. The memory size of the start point corresponds to a benchmark backbone, *i.e.*, ResNet32. The memory size of the endpoint corresponds to the model size of model-based methods using the same backbone, *i.e.*, saving 10 ResNet32 backbones in this setting. For other points of the X-axis, we can easily extend exemplar-based methods to these memory sizes by adding exemplars of equal size. For example, saving a ResNet32 model costs $463,504$ parameters (float), while saving a CIFAR image costs $3 \times 32 \times 32$ integer numbers (int). The budget of saving a backbone is equal to saving $463,504$ floats $\times 4$ bytes/float $\div (3 \times 32 \times 32)$ bytes/image $\approx 603$ instances for CIFAR. We cannot use the same backbone as the exemplar-based methods for model-based ones when the memory size is small. Hence, we divide the model parameters into ten equal parts (since there are ten incremental tasks in this setting) and look for a backbone with similar parameter numbers. For example, we separately use ConvNet, ResNet14, ResNet20, and ResNet26 as the backbone for these methods to match different memory scales. Please refer to supplementary for more details.

Given these curves, there are two questions. First, **what is a good performance measure considering memory cost?** We argue that a good CIL method with *extendability* should work for any memory cost. As a result, it is intuitive to measure the area under the curve (AUC) for these methods for a holistic overview. Observing that model-based methods work at large memory costs while failing at small costs in Fig-

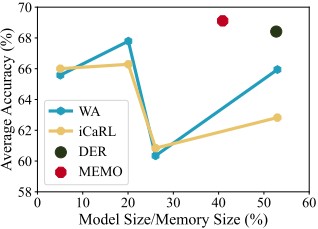
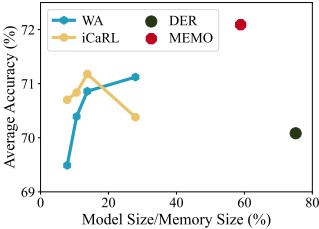

(a) Memory Size = 12.4MB          (b) Memory Size = 23.5MB

Figure 2: Performance of different methods when fixing the total budget and varying the ratio of model size to total memory size on CIFAR100.

ure 1, we argue that measuring performance at the start point and endpoint can also be representative measures. Secondly, given *a specific memory cost*, should we use **a larger model or more exemplars?** Denote the ratio of model size as $\rho = \text{Size(Model)}/\text{Size(Total)}$. We vary this ratio

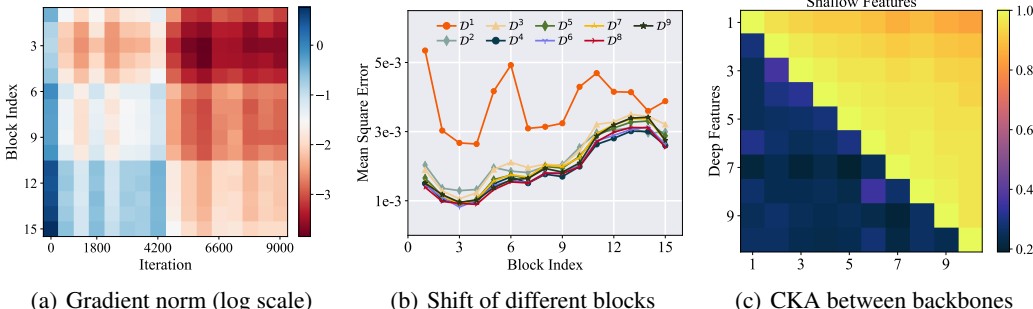

(a) Gradient norm (log scale)     (b) Shift of different blocks     (c) CKA between backbones

Figure 3: **Left**: gradient norm of different residual blocks when optimizing Eq. 1. Deeper layers have larger gradients, while shallow layers have small gradients. **Middle**: Shift between the first and last epoch of different residual blocks. Deeper layers change more, while shallow layers change less. **Right**: feature similarity (CKA) of different backbones learned by Eq. 2. The lower triangular matrix denotes the similarity between deeper layers; the upper triangular matrix denotes the similarity between shallow layers.

for exemplar-based methods at two different memory scales in Figure 2. We switch ResNet32 to ResNet44/56/110 to enlarge $\rho$. Results indicate that the benefit from exemplars shall converge when the budget is large enough (*i.e.*, Figure 2(b)), where switching to a larger model is more memory-efficient. However, this trend does not hold when the total memory budget is limited, *i.e.*, Figure 2(a), and there is no consistent rule in these figures. As a result, we report results with the same benchmark backbone in Figure 1 for consistency.

### 4.3 DO WE NEED A NEW BACKBONE PER TASK?

Aggregating the features from multiple stages can obtain diverse representations while sacrificing the memory size for storing backbones. From the memory-efficient perspective, we wonder *if all layers are equal* in CIL — if the storage of some layers is unnecessary, switching them for exemplars will be more memory-efficient for the final performance. In detail, we analyze from three aspects, *i.e.*, block-wise gradient, shift, and similarity on CIFAR100 with ResNet32. Experiments with other backbones (*e.g.*, ViT) and datasets (*e.g.*, NLP) can be found in Section C.10.

**Block-Wise Gradient:** We first conduct experiments to analyze the gradient of different residual blocks when optimizing Eq. 1. We show the gradient norm of different layers in a single task in Figure 3(a). A larger block index corresponds to deeper layers. It indicates that the gradients of shallow layers are much smaller than deep layers. As a result, deeper layers shall face stronger adjustment within new tasks, while shallow layers tend to stay unchanged during CIL.

**Block-Wise Shift:** To quantitatively measure the adjustment of different layers, we calculate the mean square error (MSE) per block between the first and last epoch for every incremental stage in Figure 3(b). It shows that the shift of deeper layers is much higher than shallow layers, indicating that deeper layers change more while shallow layers change less. It should be noted that MSE in the first task $\mathcal{D}^1$ is calculated for the randomly initialized network, which shows different trends than others. These results are consistent with the observations of gradients in Figure 3(a).

**Feature Similarity:** Observations above imply the differences between shallow and deep layers in CIL. We also train a model with Eq. 2 for 10 incremental stages, resulting in 10 backbones. We use centered kernel alignment (CKA) (Kornblith et al., 2019), an effective tool to measure the similarity of network representations to evaluate the relationships between these backbones. We can get corresponding feature maps by feeding the same batch of instances into these backbones. Afterward, CKA is applied to measure the similarity between these feature maps. We separately calculate the similarity for deep (*i.e.*, residual block 15) and shallow features (*i.e.*, residual block 5) and report the similarity matrix in Figure 3(c). The similarities between deep features are shown in the *lower triangular matrix*, and the similarities between shallow features are shown in the *upper triangular matrix*. Results indicate that the features of shallow layers among all backbones are *highly similar*, while diverse for deeper layers.

To summarize, we empirically find that *not all layers are equal in CIL*, where shallow layers yield higher similarities than deeper layers. A possible reason is that *shallow layers tend to provide general-purpose representations, whereas later layers specialize* (Maennel et al., 2020; Ansuini et al., 2019; Arpit et al., 2017; Yosinski et al., 2014; Zhang et al., 2019). Hence, expanding general layers would be less effective since they are highly similar. On the other hand, expanding and saving the specialized features is essential, which helps extract diverse representations continually.

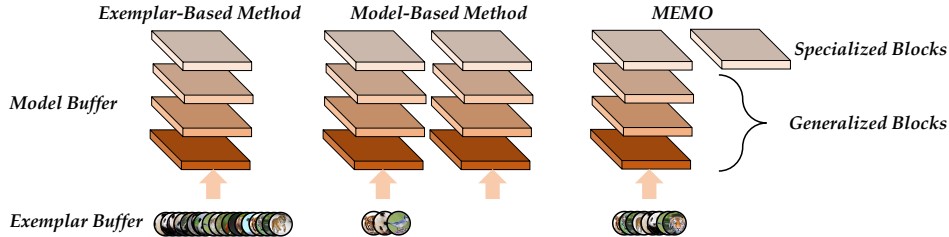

Figure 4: An overview of three typical methods. **Left**: Exemplar-based methods train a single model. **Middle**: Model-based methods train a new model per new task. **Right**: MEMO trains a new specialized block per new task. When aligning the memory cost of these methods, exemplar-based methods can save the most exemplars, while model-based methods have the least. MEMO strikes a trade-off between exemplar and model buffer.

### 4.4 MEMO: MEMORY-EFFICIENT EXPANDABLE MODEL

Motivated by the observations above, we seek to simultaneously consider saving exemplars and model extension in a memory-efficient manner. We ask:

> Given *the same memory budget*, if we share the generalized blocks and only *extend specialized blocks* for new tasks, can we further improve the performance?

Concretely, we redefine the model structure in Eq. 2 by decomposing the embedding module into *specialized and generalized* blocks, *i.e.*, $\phi(\mathbf{x}) = \phi_s(\phi_g(\mathbf{x}))$.[1] Specialized block $\phi_s(\cdot)$ corresponds to the deep layers in the network (the last basic layer in our setting, see Section B.1 for details), while generalized block $\phi_g(\cdot)$ corresponds to the rest shallow layers. We argue that the features of shallow layers can be shared across different incremental stages, *i.e.*, there is no need to create an extra $\phi_g(\cdot)$ for every new task. To this end, we can extend the feature representation by only creating specialized blocks $\phi_s$ based on shared generalized representations. We can modify the loss function in Eq. 2 into:

$$\mathcal{L}(\mathbf{x}, y) = \sum_{k=1}^{|\mathcal{Y}_b|} -\mathbb{I}(y = k) \log \mathcal{S}_k(W_{new}^\top [\phi_{s_{old}}(\phi_g(\mathbf{x})), \phi_{s_{new}}(\phi_g(\mathbf{x}))]) . \tag{3}$$

**Effect of block sharing:** We illustrate the framework of MEMO in Figure 4. There are two advantages of MEMO. Firstly, it enables a model to extract new features by adding specialized blocks continually. Hence, we can get a holistic view of the instances from various perspectives, which in turn facilitates classification. Secondly, it saves the total memory budget by sharing the generalized blocks compared to Eq. 2. It is less effective to sacrifice an extra memory budget to extract similar feature maps of these homogeneous features. Since only the last basic block is created for new tasks, we can exchange the saved budget of generalized blocks for an equal size of exemplars. In other words, *these extra exemplars will facilitate model training more than creating those generalized blocks.*

**Which Block Should be Frozen?** By comparing Eq. 2 to Eq. 3, we can find that the network structure is decomposed, and only the specialized blocks are extended. It should be noted that the old backbone is fixed when learning new tasks in Eq. 2. However, since the generalized blocks are shared during the learning stages, should they be fixed as in Eq. 2 or be dynamic to learn new classes? We conduct corresponding experiments on CIFAR100 and report the

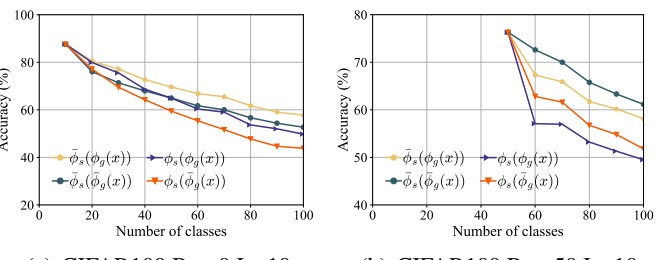

(a) CIFAR100 Base0 Inc10      (b) CIFAR100 Base50 Inc10

Figure 5: Experiments about specialized and generalized blocks. Specialized blocks should be fixed; while fixing or not generalized blocks depends on the number of classes in the base stage. Block with $\bar{\phi}$ means frozen, while without a bar means trainable.

incremental performance by changing the learnable blocks in Figure 5. In detail, we fix/unfix the specialized and generalized blocks in Eq. 3 when learning new tasks, which yields four combinations. We separately evaluate the results in two settings by changing the number of base classes. As shown in Figure 5, there are two core observations. Firstly, by comparing the results of whether freezing the

---

[1]We illustrate the implementation with ResNet, which can also be applied to other network structures like VGGNet and Inception. See Section B.2 for details.

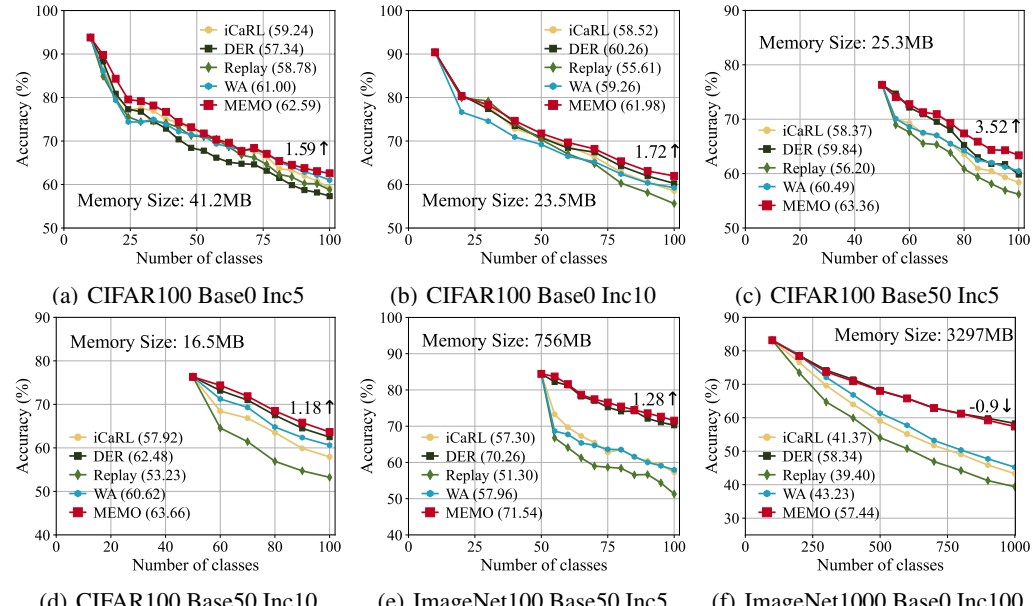

(a) CIFAR100 Base0 Inc5    (b) CIFAR100 Base0 Inc10    (c) CIFAR100 Base50 Inc5

(d) CIFAR100 Base50 Inc10    (e) ImageNet100 Base50 Inc5    (f) ImageNet1000 Base0 Inc100

Figure 6: Top-1 accuracy along incremental stages. All methods are compared under the same memory budget as denoted in the image (aligned with DER). We report the performance gap after the last task of MEMO and the runner-up method at the end of the line. We list the last accuracy of each method in the legend, and report the incremental accuracy and configurations in Section A.3.

Table 1: Memory-aware performance measures for CIL. AUC depicts the dynamic ability with the change of memory size, and APM depicts the capacity at some specific memory cost.

| CIFAR100 | AUC-A | AUC-L | APM-S | APM-E |
|---|---|---|---|---|
| Replay | 10.49 | 8.02 | 7.68 | 2.97 |
| iCaRL | 10.81 | 8.64 | 8.32 | 3.00 |
| WA | 10.80 | 8.92 | **8.57** | 2.95 |
| DER | 10.74 | 8.95 | 7.05 | 2.97 |
| MEMO | **10.85** | **9.03** | 7.18 | **3.06** |

| ImageNet100 | AUC-A | AUC-L | APM-S | APM-E |
|---|---|---|---|---|
| Replay | 553.6 | 470.1 | 0.137 | 5.2e-2 |
| iCaRL | 607.1 | 527.5 | 0.164 | 5.4e-2 |
| WA | 666.0 | 581.7 | 0.195 | 5.8e-2 |
| DER | 699.0 | 639.1 | 0.192 | 5.8e-2 |
| MEMO | **713.0** | **654.6** | **0.196** | **6.1e-2** |

specialized blocks, we can tell that methods freezing the specialized blocks of former tasks have stronger performance than those do not freeze specialized blocks. It indicates that the **specialized blocks of former tasks should be frozen to obtain diverse feature representations**. Secondly, when the base classes are limited (*e.g.*, 10 classes), the generalized blocks are not generalizable and transferable enough to capture the feature representations, which need to be incrementally updated. By contrast, vast base classes (*e.g.*, 50 classes) can build transferable generalized blocks, and freezing them can get better performance under such a setting. See Section C.2 for more results.

## 5    EXPERIMENT

### 5.1    REVISITING BENCHMARK COMPARISON FOR CIL

Former works evaluate the performance with the accuracy trend along incremental stages, which lack consideration of the memory budget and *compare models at different X coordinates*. As a result, in this section, we strike a balance between different methods by aligning the memory cost of different methods to the endpoint in Figure 1 (which is also the memory cost of DER). We show 6 typical results in Figure 6, containing different settings discussed in Section 4.1 on three benchmark datasets. We summarize two main conclusions from these figures. Firstly, the improvement of DER over other methods is not so much as reported in the original paper, which outperforms others substantially by 10% or more. In our observation, the *improvement of DER than others under the fair comparison is much less*, indicating that saving models shows slightly greater potential than saving exemplars when the memory budget is large. Secondly, MEMO outperforms DER by a substantial margin in most

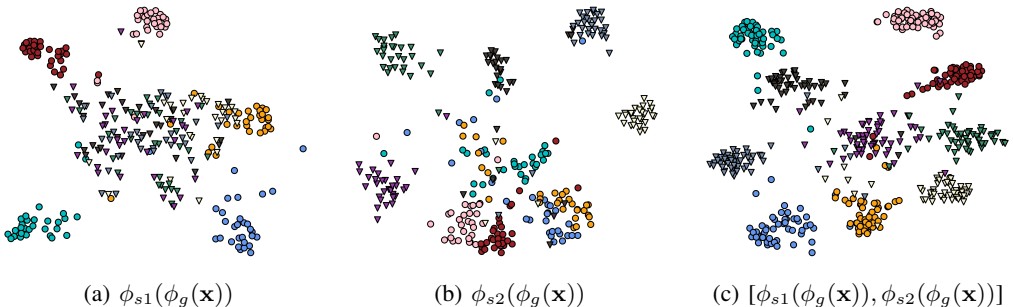

(a) $\phi_{s1}(\phi_g(\mathbf{x}))$       (b) $\phi_{s2}(\phi_g(\mathbf{x}))$       (c) $[\phi_{s1}(\phi_g(\mathbf{x})), \phi_{s2}(\phi_g(\mathbf{x}))]$

Figure 7: t-SNE visualizations on CIFAR100 of different specialized blocks learned by MEMO. Classes 1-5 are shown in dots, and classes 6-10 are shown in triangles.

cases, indicating ours is a simple yet effective way to organize CIL models with memory efficiency. These conclusions are consistent with our observations in Figure 2 and 3.

## 5.2 HOW TO MEASURE THE PERFORMANCE OF CIL MODELS HOLISTICALLY?

As discussed in Section 4.2, a suitable CIL model should be able to handle the task with *any* memory budget. Changing the model size from small to large enables us to evaluate different methods *holistically*. We observe that all the methods benefit from the increased memory size and perform better as it becomes larger in Figure 1. Hence, new *performance measures* should be proposed considering the model capacity. We first suggest the area under the performance-memory curve (AUC) since the curve of each method indicates the dynamic ability with the change of model size. We calculate **AUC-A** and **AUC-L**, standing for the AUC under the average performance-memory curve and last performance-memory curve. Similarly, we find that the intersection usually emerges near the start point of Figure 1. As a result, we can also calculate the accuracy per model size at the start point and endpoint, denoted as **APM-S** and **APM-E** separately. They represent the performance of the algorithm at different memory scales. It should be noted that these measures are not normalized into the centesimal scale, and the ranking among different methods is more important than the relative values. We report these measures in Table 1, where MEMO obtains the best performance in most cases (7 out of 8). Since all the methods are compared under the same budget, MEMO achieves the performance improvement *for free*, verifying its memory efficiency.

## 5.3 WHAT IS LEARNED BY SPECIALIZED BLOCKS?

Adding the specialized blocks helps extract diverse feature representations of a single instance. In this section, we train our model on CIFAR100 between two incremental stages; each contains five classes. We use t-SNE (Van der Maaten & Hinton, 2008) to visualize the property of these blocks. Classes from the first task are shown in dots, and classes from the second task are shown in triangles. We visualize the learned embedding of these separate specialized blocks in Figure 7(a) and 7(b). We can infer that the specialized blocks are optimized to discriminate the corresponding task, *i.e.*, $\phi_{s1}(\phi_g(\mathbf{x}))$ can recognize classes 1∼5 clearly, and $\phi_{s2}(\phi_g(\mathbf{x}))$ can tackle classes 6∼10 easily. When we aggregate the embeddings from these two backbones, *i.e.*, $[\phi_{s1}(\phi_g(\mathbf{x})), \phi_{s2}(\phi_g(\mathbf{x}))]$, the concatenated features are able to capture all the classes seen before. Results indicate that specialized blocks, which are fixed after learning the corresponding task, act as 'unforgettable checkpoints.' They will not lose discrimination as data evolves. Hence, we can aggregate diverse feature representations in the aggregated high dimension and divide decision boundaries easily.

## 6 CONCLUSION

Class-incremental learning ability is of great importance to real-world learning systems, requiring a model to learn new classes without forgetting old ones. In this paper, we answer two questions in CIL. Firstly, we fairly compare different methods by aligning the memory size at the same scale. Secondly, we find that not all layers are needed to be created and stored for new tasks, and propose a simple yet effective baseline, obtaining state-of-the-art performance *for free* in the fair comparison. Experiments verify the memory efficiency of our proposed method.

**Limitations:** CIL methods can also be divided by whether using extra memory. Apart from the methods discussed in this paper, there are other methods that do not save exemplars or models. We only concentrate on the methods with extra memory and select several typical methods for evaluation.

ACKNOWLEDGMENT

This work is partially supported by NSFC (61921006, 62006112, 62250069), NSF of Jiangsu Province (BK20200313), Collaborative Innovation Center of Novel Software Technology and Industrialization, China Scholarship Council (CSC202206190134).

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

# Supplementary Material

Class-incremental learning (CIL) is of great importance to the machine learning community. In the main paper, we answer two questions in CIL. The first is about how to fairly compare different methods, and we achieve this goal by aligning the memory cost. The second is about organizing the model with memory efficiency, and our proposed MEMO maintains the diverse feature representations with a modest memory cost. In the supplementary, we report more details about the experimental results mentioned in the main paper. We also provide more empirical evaluations and discussions. The supplementary material is organized according to the two questions above — we first supply details to fairly compare different methods and then discuss how to manage the model in a memory-efficient manner. Afterward, we give the additional experimental and implementation details.

- Section A reports the implementation details of the models and exemplars in the performance-memory curve of the main paper, and numerical results in the benchmark comparison;

- Section B discusses the variations of MEMO, including the definition of specialized and generalized blocks and other deep network structures;

- Section C provides extra illustrative experimental evaluations that cannot be included in the main paper due to page limit, including the last accuracy-memory curve, gradient norms for all tasks, CKA visualizations of different blocks, the ablations about which layer to freeze, running time comparison, and CIL performance with multiple runs;

- Section D holistically discusses the implementations of related work and ours, the choice of compared methods, and broader impacts.

## A  IMPLEMENTATION DETAILS OF PERFORMANCE-MEMORY CURVE

We give the performance-memory curve in the main paper as one main contribution to fairly comparing different CIL methods. In this section, we give the detailed implementations of each point on the X coordinate. We will start with CIFAR100 (Krizhevsky et al., 2009), and then discuss ImageNet100 (Deng et al., 2009).

In the following discussions, we use $\mathcal{E}$ to represent the exemplar set. $|\mathcal{E}|$ denotes the number of exemplars, and $S(\mathcal{E})$ represents the memory size (in MB) that saving these exemplars consume. Following the benchmark implementation (Rebuffi et al., 2017), 2,000 exemplars are saved for every method for CIFAR100 and ImageNet100. Hence, the exemplar size of each method is denoted as $|\mathcal{E}| = 2000 + E$, where $E$ corresponds to the extra exemplars exchanged from the model size, as discussed in the main paper. We use '# Parameters' to represent the number of parameters and 'Model Size' to represent the memory budget (in MB) it costs to save this model in memory. The total memory size (*i.e.*, numbers on the X coordinate) is the sum of exemplars and the models.

Since iCaRL (Rebuffi et al., 2017), Replay (Chaudhry et al., 2019) and WA (Zhao et al., 2020) are typical exemplar-based methods, they use the same network backbone with the same model size. Hence, they have equal memory sizes. DER (Yan et al., 2021) sacrifices the memory size to store the backbone from history, and it has the least exemplars. Compared to DER, MEMO does not keep the duplicated generalized blocks from history and saves much memory size to change into exemplars.

In the following discussions, we first give the tables to illustrate the implementation of different methods and report their incremental performance with figures. We report the improvement of MEMO against the runner-up method at the end of each line in the figures and analyze the empirical evaluations after the tables and figures.

### A.1  IMPLEMENTATIONS OF CIFAR100

There are five X coordinates in the curve of CIFAR100, *e.g.*, 7.6, 12.4, 16.0, 19.8, and 23.5 MB. Following, we show the detailed implementation of different methods at these scales.

| 7.6MB | $|\mathcal{E}|$ | $S(\mathcal{E})$ | Model Type | # Parameters | Model Size |
|---|---|---|---|---|---|
| Replay | 2000 | 5.85MB | ResNet32 | 0.46M | 1.76MB |
| iCaRL | 2000 | 5.85MB | ResNet32 | 0.46M | 1.76MB |
| WA | 2000 | 5.85MB | ResNet32 | 0.46M | 1.76MB |
| DER | 2096 | 6.14MB | ConvNet2 | 0.38M | 1.48MB |
| MEMO | 2118 | 6.20MB | ConvNet2 | 0.37M | 1.42MB |

Table 2: Implementation details when memory size= 7.6 MB

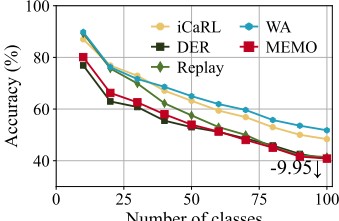

Figure 8: CIL performance

**CIFAR100 with 7.6 MB Memory Size:** The implementations are shown in Table 2 and Figure 8. 7.6 MB is a relatively small memory size. Since we need to align the total budget of these methods, we are only able to use small backbones for DER and MEMO. These small backbones, *i.e.*, ConvNet with two convolutional layers, has much fewer parameters than ResNet32, and saving 10 ConvNets matches the memory size of a single ResNet32 (1.48MB versus 1.76MB). We can infer from the table that DER and MEMO are restricted by the learning ability of the inferior backbones, which perform poorly in the base session. These results are consistent with the conclusions in the main paper that model-based methods are inferior to exemplar-based methods with a small memory budget.

| 12.4MB | $|\mathcal{E}|$ | $S(\mathcal{E})$ | Model Type | # Parameters | Model Size |
|---|---|---|---|---|---|
| Replay | 3634 | 10.64MB | ResNet32 | 0.46M | 1.76MB |
| iCaRL | 3634 | 10.64MB | ResNet32 | 0.46M | 1.76MB |
| WA | 3634 | 10.64MB | ResNet32 | 0.46M | 1.76MB |
| DER | 2000 | 5.85MB | ResNet14 | 1.70M | 6.55MB |
| MEMO | 2495 | 7.32MB | ResNet14 | 1.33M | 5.10MB |

Table 3: Implementation details when memory size= 12.4 MB

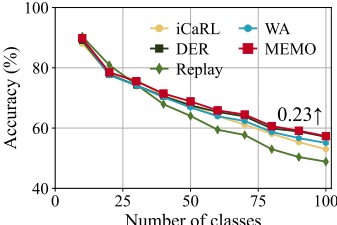

Figure 9: CIL performance

**CIFAR100 with 12.4 MB Memory Size:** The implementations are shown in Table 3 and Figure 9. By raising the total memory cost to 12.4 MB, exemplar-based methods can utilize the extra memory size to exchange 1634 exemplars, and model-based methods can switch to more powerful backbones to get better representation ability. We use ResNet14 for DER and MEMO in this setting. We can infer that model-based methods show competitive results with stronger backbones and outperform exemplar-based methods in this setting. These results are consistent with the conclusions in the main paper that the *intersection* between these two groups of methods exists near the start point.

| 16.0MB | $|\mathcal{E}|$ | $S(\mathcal{E})$ | Model Type | # Parameters | Model Size |
|---|---|---|---|---|---|
| Replay | 4900 | 14.3MB | ResNet32 | 0.46M | 1.76MB |
| iCaRL | 4900 | 14.3MB | ResNet32 | 0.46M | 1.76MB |
| WA | 4900 | 14.3MB | ResNet32 | 0.46M | 1.76MB |
| DER | 2000 | 5.85MB | ResNet20 | 2.69M | 10.2MB |
| MEMO | 2768 | 8.10MB | ResNet20 | 2.1M | 8.01MB |

Table 4: Implementation details when memory size= 16.0 MB

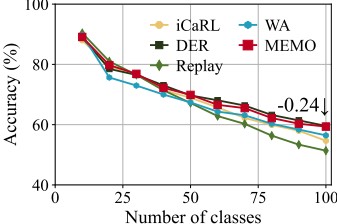

Figure 10: CIL performance

**CIFAR100 with 16.0 MB Memory Size:** The implementations are shown in Table 4 and Figure 10. By raising the total memory cost to 16.0 MB, exemplar-based methods can utilize the extra memory size to exchange 2900 exemplars, and model-based methods can switch to larger backbones to get better representation ability. We use ResNet20 for DER and MEMO in this setting. The results are consistent with the former setting, where we can infer that model-based methods show competitive results with stronger backbones and outperform exemplar-based methods.

| 19.8MB | $|\mathcal{E}|$ | $S(\mathcal{E})$ | Model Type | # Parameters | Model Size |
|---|---|---|---|---|---|
| Replay | 6165 | 18.06MB | ResNet32 | 0.46M | 1.76MB |
| iCaRL | 6165 | 18.06MB | ResNet32 | 0.46M | 1.76MB |
| WA | 6165 | 18.06MB | ResNet32 | 0.46M | 1.76MB |
| DER | 2000 | 5.85MB | ResNet26 | 3.60M | 13.9MB |
| MEMO | 3040 | 8.91MB | ResNet26 | 2.86M | 10.92MB |

Table 5: Implementation details when memory size= 19.8 MB

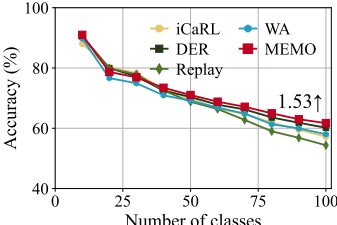

Figure 11: CIL performance

**CIFAR100 with 19.8 MB Memory Size:** The implementations are shown in Table 5 and Figure 11. By raising the total memory cost to 19.8 MB, exemplar-based methods can utilize the extra memory size to exchange 4165 exemplars, and model-based methods can switch to larger backbones to get better representation ability. We use ResNet26 for DER and MEMO in this setting. The results are consistent with the former setting, where we can infer that model-based methods show competitive results with stronger backbones and outperform exemplar-based methods.

| 23.5MB | $|\mathcal{E}|$ | $S(\mathcal{E})$ | Model Type | # Parameters | Model Size |
|---|---|---|---|---|---|
| Replay | 7431 | 21.76MB | ResNet32 | 0.46M | 1.75MB |
| iCaRL | 7431 | 21.76MB | ResNet32 | 0.46M | 1.75MB |
| WA | 7431 | 21.76MB | ResNet32 | 0.46M | 1.75MB |
| DER | 2000 | 5.86MB | ResNet32 | 4.63M | 17.68MB |
| MEMO | 3312 | 9.7MB | ResNet32 | 3.62M | 13.83MB |

Table 6: Implementation details when memory size= 23.5 MB

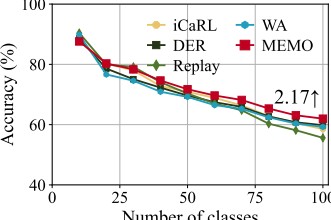

Figure 12: CIL performance

**CIFAR100 with 23.5 MB Memory Size:** The implementations are shown in Table 6 and Figure 12. By raising the total memory cost to 23.5 MB, exemplar-based methods can utilize the extra memory size to exchange 5431 exemplars, and model-based methods can switch to larger backbones to get better representation ability. We use ResNet32 for DER and MEMO in this setting. The results are consistent with the conclusions in Section 5.1. Model-based methods are better than exemplar-based methods with large memory sizes, but the performance gap is not so large as reported in (Yan et al., 2021) when fairly compared.

To summarize, we conduct fair comparisons for exemplar-based and model-based methods by varying the memory size from small to large. Results indicate that exemplar-based methods are competitive with small memory sizes, while model-based methods are competitive with large ones. Our proposed MEMO obtains the best performance in most cases in these settings.

### A.2 IMPLEMENTATIONS OF IMAGENET100

Similar to CIFAR100, we can conduct an exchange between the model and exemplars on ImageNet100. For example, saving a ResNet18 model costs $11,176,512$ parameters (float), while saving an ImageNet image costs $3 \times 224 \times 224$ integer numbers (int). The budget of saving a backbone is equal to saving $11,176,512$ floats $\times 4$ bytes/float $\div (3 \times 224 \times 224)$ bytes/image $\approx 297$ images for ImageNet. We conduct the experiment with ImageNet100, Base50 Inc5, as discussed in the main paper. There are six X coordinates in the curve of ImageNet100, *e.g.*, 329, 493, 755, 872, 1180 and 1273 MB. Following, we show the detailed implementation of different methods at these scales.

| 329MB | $|\mathcal{E}|$ | $S(\mathcal{E})$ | Model Type | # Parameters | Model Size |
|---|---|---|---|---|---|
| Replay | 2000 | 287MB | ResNet18 | 11.17M | 42.6MB |
| iCaRL | 2000 | 287MB | ResNet18 | 11.17M | 42.6MB |
| WA | 2000 | 287MB | ResNet18 | 11.17M | 42.6MB |
| DER | 2032 | 291MB | ConvNet4 | 9.96M | 38.0MB |
| MEMO | 2115 | 303MB | ConvNet4 | 6.81M | 26.0MB |

Table 7: Implementation details when memory size=329MB

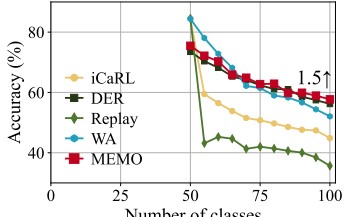

Figure 13: CIL performance

**ImageNet100 with 329 MB Memory Size:** The implementations are shown in Table 7 and Figure 13. 329 MB is a relatively small memory size. Since we need to align the total budget of these methods, we are only able to use small backbones for DER and MEMO. These small backbones, *i.e.*, ConvNet with four convolutional layers, has much fewer parameters than ResNet18, and saving 10 ConvNets matches the memory size of a single ResNet18. We can infer from the table that DER and MEMO are restricted by the learning ability of the inferior backbones, which perform poorly in the base session. However, our proposed MEMO outperforms these better backbones by saving the 'unforgettable checkpoints,' which obtains the best last accuracy and average accuracy in this case.

| 493MB | $|\mathcal{E}|$ | $S(\mathcal{E})$ | Model Type | # Parameters | Model Size |
|---|---|---|---|---|---|
| Replay | 3136 | 450MB | ResNet18 | 11.17M | 42.6MB |
| iCaRL | 3136 | 450MB | ResNet18 | 11.17M | 42.6MB |
| WA | 3136 | 450MB | ResNet18 | 11.17M | 42.6MB |
| DER | 2000 | 287MB | ResNet10 | 53.96M | 205MB |
| MEMO | 2327 | 334MB | ResNet10 | 41.63M | 158MB |

Table 8: Implementation details when memory size=493MB

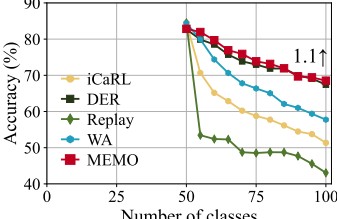

Figure 14: CIL performance

**ImageNet100 with 493 MB Memory Size:** The implementations are shown in Table 8 and Figure 14. By raising the total memory cost to 493 MB, exemplar-based methods can utilize the extra memory size to exchange 1136 exemplars, and model-based methods can switch to larger backbones to get better representation ability. We use ResNet10 for DER and MEMO in this setting. We can infer from the figure that model-based methods show competitive results with stronger backbones and outperform exemplar-based methods in this setting.

| 755MB | $|\mathcal{E}|$ | $S(\mathcal{E})$ | Model Type | # Parameters | Model Size |
|---|---|---|---|---|---|
| Replay | 4970 | 713.5MB | ResNet18 | 11.17M | 42.6MB |
| iCaRL | 4970 | 713.5MB | ResNet18 | 11.17M | 42.6MB |
| WA | 4970 | 713.5MB | ResNet18 | 11.17M | 42.6MB |
| DER | 2000 | 287MB | ResNet18 | 122.9M | 468MB |
| MEMO | 2739 | 393.2MB | ResNet18 | 95.11M | 362.8MB |

Table 9: Implementation details when memory size=755MB

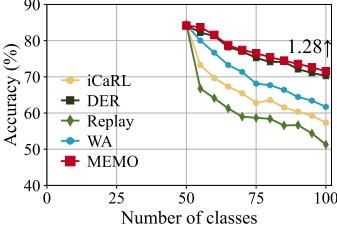

Figure 15: CIL performance

**ImageNet100 with 755 MB Memory Size:** The implementations are shown in Table 9 and Figure 15. By raising the total memory cost to 755 MB, exemplar-based methods can utilize the extra memory size to exchange 2970 exemplars, and model-based methods can switch to larger backbones to get better representation ability. We use ResNet18 for DER and MEMO in this setting. The results are consistent with the former setting, where we can infer that model-based methods show competitive results with stronger backbones and outperform exemplar-based methods.

| 872MB | $|\mathcal{E}|$ | $S(\mathcal{E})$ | Model Type | # Parameters | Model Size |
|---|---|---|---|---|---|
| Replay | 5779 | 829MB | ResNet18 | 11.17M | 42.6MB |
| iCaRL | 5779 | 829MB | ResNet18 | 11.17M | 42.6MB |
| WA | 5779 | 829MB | ResNet18 | 11.17M | 42.6MB |
| DER | 2000 | 287MB | ResNet26 | 153.4M | 585.2MB |
| MEMO | 2915 | 417MB | ResNet26 | 119.0M | 453MB |

Table 10: Implementation details when memory size=872MB

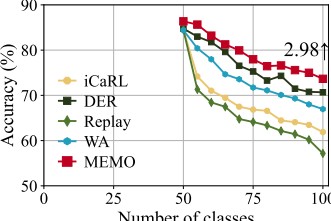

Figure 16: CIL performance

**ImageNet100 with 872 MB Memory Size:** The implementations are shown in Table 10 and Figure 16. By raising the total memory cost to 872 MB, exemplar-based methods can utilize the extra memory size to exchange 3779 exemplars, and model-based methods can switch to larger backbones for better representation ability. We use ResNet26 for DER and MEMO in this setting.

| 1180MB | $|\mathcal{E}|$ | $S(\mathcal{E})$ | Model Type | # Parameters | Model Size |
|---|---|---|---|---|---|
| Replay | 7924 | 1137MB | ResNet18 | 11.17M | 42.6MB |
| iCaRL | 7924 | 1137MB | ResNet18 | 11.17M | 42.6MB |
| WA | 7924 | 1137MB | ResNet18 | 11.17M | 42.6MB |
| DER | 2000 | 287MB | ResNet34 | 234.1M | 893MB |
| MEMO | 4170 | 598MB | ResNet34 | 152.4M | 581MB |

Table 11: Implementation details when memory size=1180MB

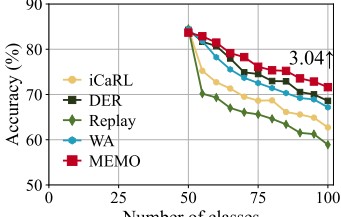

Figure 17: CIL performance

**ImageNet100 with 1180 MB Memory Size:** The implementations are shown in Table 11 and Figure 17. By raising the total memory cost to 1180 MB, exemplar-based methods can utilize the extra memory size to exchange 5924 exemplars, and model-based methods can switch to larger backbones to get better representation ability. We use ResNet34 for DER and MEMO in this setting.

| 1273MB | $|\mathcal{E}|$ | $S(\mathcal{E})$ | Model Type | # Parameters | Model Size |
|---|---|---|---|---|---|
| Replay | 8574 | 1230MB | ResNet18 | 11.17M | 42.6MB |
| iCaRL | 8574 | 1230MB | ResNet18 | 11.17M | 42.6MB |
| WA | 8574 | 1230MB | ResNet18 | 11.17M | 42.6MB |
| DER | 2000 | 287MB | ResNet50 | 258.6M | 986MB |
| MEMO | 4270 | 612MB | ResNet50 | 173.2M | 660MB |

Table 12: Implementation details when memory size=1273MB

Figure 18: CIL performance

**ImageNet100 with 1273 MB Memory Size:** The implementations are shown in Table 12 and Figure 18. By raising the total memory cost to 1273 MB, exemplar-based methods can utilize the extra memory size to exchange 6574 exemplars, and model-based methods can switch to larger backbones to get better representation ability. We use ResNet50 for DER and MEMO in this setting. The results are consistent with the conclusions in Section 5.1. Model-based methods are better than exemplar-based methods with large memory sizes, but the performance gap is not so large as reported in (Yan et al., 2021) when fairly compared.

**Discussion about Backbones:** It should be noted that ResNet18 is the benchmark backbone for ImageNet, and the memory size for 872, 1180, and 1273 MB are larger than the benchmark setting. We conduct these experiments for two reasons. First, handling large-scale image inputs require more convolutional layers, and it is hard to find typical models with small memory budgets. Second, we would like to investigate the performance when the model is large enough to see whether the improvement of stronger backbones will converge, and the results successfully verify our assumptions.

To summarize, we conduct fair comparisons for exemplar-based and model-based methods by varying the memory size from small to large. Results indicate that exemplar-based methods are competitive with small memory sizes, while model-based methods are competitive with large ones. Our proposed MEMO obtains the best performance in most cases in these settings.

### A.3 Numerical Results and configurations for Section 5.1

In this section, we give the numerical incremental performance of different methods and configurations in Section 5.1. We first list the incremental and average accuracy in Table 13, 14, 15, 16,17, 18, and give the setting in Table 19, 20, 21, 22, 23, 24.

As we discussed in the main paper, a fair comparison among different methods should be aligned to the same memory budget. Since DER (Yan et al., 2021) requires saving multiple backbones, training DER requires the largest memory budget. Hence, we align the budget of other methods to DER (as shown in Table 19, 20, 21, 22, 23, 24) by saving more exemplars for them.

Table 13: Incremental and average accuracy comparison of different methods under CIFAR100 Base0 Inc5 setting.

| Method | Accuracy in each session (%) ↑ | | | | | | | | | | | | | | | | | | | | Average |
|--------|------|------|------|------|------|------|------|------|------|------|------|------|------|------|------|------|------|------|------|------|---------|
| | 1 | 2 | 3 | 4 | 5 | 6 | 7 | 8 | 9 | 10 | 11 | 12 | 13 | 14 | 15 | 16 | 17 | 18 | 19 | 20 | |
| Replay | 93.80 | 84.90 | 79.67 | 75.55 | 74.44 | 74.57 | 74.14 | 73.12 | 71.22 | 70.96 | 70.42 | 68.63 | 66.75 | 66.19 | 64.63 | 62.34 | 61.73 | 60.24 | 60.17 | 58.78 | 70.61 |
| iCaRL | 93.80 | 86.70 | 80.67 | 77.15 | 77.52 | 76.97 | 75.03 | 74.22 | 72.62 | 71.90 | 70.44 | 69.12 | 68.09 | 67.59 | 66.39 | 63.62 | 63.28 | 61.98 | 60.81 | 59.24 | 71.85 |
| WA | 93.80 | 86.10 | 79.40 | 74.40 | 74.36 | 74.97 | 73.80 | 72.18 | 71.49 | 71.10 | 69.35 | 68.65 | 68.00 | 67.97 | 67.08 | 65.20 | 64.04 | 62.89 | 62.14 | 61.00 | 71.39 |
| DER | 93.80 | 88.30 | 80.87 | 77.30 | 76.76 | 74.53 | 72.86 | 70.38 | 68.44 | 67.78 | 66.18 | 65.13 | 64.78 | 64.63 | 63.19 | 61.52 | 59.86 | 58.73 | 58.17 | 57.34 | 69.52 |
| MEMO | 93.80 | 89.80 | 84.33 | 79.55 | 79.16 | 78.17 | 76.71 | 74.38 | 73.16 | 71.74 | 70.33 | 69.62 | 67.68 | 68.41 | 67.07 | 65.41 | 64.53 | 63.78 | 63.07 | 62.59 | 73.16 |

Table 14: Incremental and average accuracy comparison of different methods under CIFAR100 Base0 Inc10 setting.

| Method | Accuracy in each session (%) ↑ | | | | | | | | | | Average |
|--------|-------|-------|-------|-------|-------|-------|-------|-------|-------|-------|---------|
| | 1 | 2 | 3 | 4 | 5 | 6 | 7 | 8 | 9 | 10 | |
| Replay | 90.40 | 79.90 | 79.20 | 74.08 | 70.32 | 67.03 | 64.76 | 60.31 | 58.14 | 55.61 | 69.98 |
| iCaRL | 90.40 | 80.25 | 78.00 | 72.80 | 70.88 | 68.88 | 66.37 | 62.82 | 60.51 | 58.52 | 70.94 |
| WA | 90.40 | 76.65 | 74.57 | 70.92 | 69.20 | 66.48 | 65.24 | 62.41 | 60.37 | 59.26 | 69.55 |
| DER | 90.40 | 80.55 | 77.37 | 73.47 | 70.78 | 68.38 | 67.43 | 64.22 | 61.93 | 60.26 | 71.47 |
| MEMO | 90.40 | 80.30 | 78.33 | 74.65 | 71.74 | 69.67 | 68.19 | 65.34 | 63.10 | 61.98 | 72.37 |

Table 15: Incremental and average accuracy comparison of different methods under CIFAR100 Base50 Inc5 setting.

| Method | Accuracy in each session (%) ↑ | | | | | | | | | | | Average |
|--------|-------|-------|-------|-------|-------|-------|-------|-------|-------|-------|-------|---------|
| | 1 | 2 | 3 | 4 | 5 | 6 | 7 | 8 | 9 | 10 | 11 | |
| Replay | 76.32 | 68.96 | 67.60 | 65.55 | 65.34 | 63.84 | 60.86 | 59.39 | 58.12 | 56.93 | 56.20 | 63.55 |
| iCaRL | 76.32 | 70.00 | 69.28 | 67.38 | 67.00 | 65.45 | 63.55 | 60.95 | 60.47 | 59.31 | 58.37 | 65.28 |
| WA | 76.32 | 70.09 | 68.58 | 67.54 | 67.04 | 65.52 | 64.24 | 62.49 | 62.00 | 61.21 | 60.49 | 65.95 |
| DER | 76.32 | 74.71 | 72.15 | 70.92 | 69.54 | 68.03 | 65.26 | 63.00 | 61.80 | 61.73 | 59.84 | 67.57 |
| MEMO | 76.32 | 74.05 | 72.77 | 71.28 | 70.93 | 69.29 | 67.38 | 65.88 | 64.36 | 64.29 | 63.36 | 69.08 |

Table 16: Incremental and average accuracy comparison of different methods under CIFAR100 Base50 Inc10 setting.

| Method | Accuracy in each session (%) ↑ | | | | | | Average |
|--------|-------|-------|-------|-------|-------|-------|---------|
| | 1 | 2 | 3 | 4 | 5 | 6 | |
| Replay | 76.32 | 64.57 | 61.43 | 56.91 | 54.72 | 53.23 | 61.19 |
| iCaRL | 76.32 | 68.43 | 66.83 | 63.52 | 59.89 | 57.92 | 65.48 |
| WA | 76.32 | 71.25 | 69.30 | 64.79 | 62.37 | 60.62 | 67.44 |
| DER | 76.32 | 73.17 | 70.99 | 67.51 | 64.49 | 62.48 | 69.16 |
| MEMO | 76.32 | 74.37 | 71.87 | 68.50 | 65.81 | 63.66 | 70.09 |

Table 17: Incremental and average accuracy comparison of different methods under ImageNet100 Base50 Inc5 setting.

| Method | Accuracy in each session (%) ↑ | | | | | | | | | | | Average |
|---|---|---|---|---|---|---|---|---|---|---|---|---|
| | 1 | 2 | 3 | 4 | 5 | 6 | 7 | 8 | 9 | 10 | 11 | |
| Replay | 84.44 | 66.76 | 64.10 | 61.26 | 59.00 | 58.67 | 58.40 | 56.56 | 56.67 | 54.36 | 51.30 | 61.04 |
| iCaRL | 84.44 | 73.27 | 69.70 | 67.29 | 65.46 | 62.80 | 63.55 | 61.53 | 60.36 | 59.26 | 57.30 | 65.90 |
| WA | 84.44 | 68.65 | 67.73 | 65.38 | 64.71 | 63.63 | 63.52 | 61.60 | 60.02 | 59.09 | 57.96 | 65.15 |
| DER | 84.44 | 82.22 | 81.30 | 78.22 | 77.03 | 75.23 | 74.10 | 74.14 | 72.11 | 71.07 | 70.26 | 76.37 |
| MEMO | 84.44 | 83.71 | 81.60 | 78.68 | 77.43 | 76.53 | 75.42 | 74.49 | 73.56 | 72.63 | 71.54 | 77.27 |

Table 18: Incremental and average accuracy comparison of different methods under ImageNet1000 Base0 Inc100 setting.

| Method | Accuracy in each session (%) ↑ | | | | | | | | | | Average |
|---|---|---|---|---|---|---|---|---|---|---|---|
| | 1 | 2 | 3 | 4 | 5 | 6 | 7 | 8 | 9 | 10 | |
| Replay | 83.16 | 73.47 | 64.72 | 59.93 | 54.05 | 50.78 | 46.91 | 44.24 | 41.22 | 39.40 | 55.78 |
| iCaRL | 81.16 | 74.56 | 67.63 | 61.98 | 57.03 | 53.15 | 49.72 | 47.07 | 43.88 | 41.30 | 57.94 |
| WA | 81.16 | 76.87 | 70.09 | 64.84 | 59.38 | 55.75 | 51.22 | 48.37 | 45.72 | 43.23 | 59.86 |
| DER | 83.16 | 78.64 | 74.21 | 71.44 | 68.13 | 65.80 | 62.9 | 61.21 | 59.84 | 58.30 | 68.36 |
| MEMO | 83.16 | 78.47 | 73.72 | 70.93 | 68.05 | 65.78 | 62.91 | 61.24 | 59.22 | 57.40 | 68.09 |

Table 19: Configurations of different methods under CIFAR100 Base0 Inc5 setting.

| 41.22MB | $|\mathcal{E}|$ | $S(\mathcal{E})$ | Model Type | # Parameters | Model Size |
|---|---|---|---|---|---|
| Replay | 13466 | 39.47MB | ResNet32 | 0.46M | 1.75MB |
| iCaRL | 13466 | 39.47MB | ResNet32 | 0.46M | 1.75MB |
| WA | 13466 | 39.47MB | ResNet32 | 0.46M | 1.75MB |
| DER | 2000 | 5.86MB | ResNet32 | 9.27M | 35.36MB |
| MEMO | 4771 | 13.98MB | ResNet32 | 7.14M | 27.24MB |

Table 20: Configurations of different methods under CIFAR100 Base0 Inc10 setting.

| 23.5MB | $|\mathcal{E}|$ | $S(\mathcal{E})$ | Model Type | # Parameters | Model Size |
|---|---|---|---|---|---|
| Replay | 7431 | 21.76MB | ResNet32 | 0.46M | 1.75MB |
| iCaRL | 7431 | 21.76MB | ResNet32 | 0.46M | 1.75MB |
| WA | 7431 | 21.76MB | ResNet32 | 0.46M | 1.75MB |
| DER | 2000 | 5.86MB | ResNet32 | 4.63M | 17.68MB |
| MEMO | 3312 | 9.7MB | ResNet32 | 3.62M | 13.83MB |

Table 21: Configurations of different methods under CIFAR100 Base50 Inc5 setting.

| 25.3MB | $|\mathcal{E}|$ | $S(\mathcal{E})$ | Model Type | # Parameters | Model Size |
|---|---|---|---|---|---|
| Replay | 8035 | 23.53MB | ResNet32 | 0.46M | 1.75MB |
| iCaRL | 8035 | 23.53MB | ResNet32 | 0.46M | 1.75MB |
| WA | 8035 | 23.53MB | ResNet32 | 0.46M | 1.75MB |
| DER | 2000 | 5.86MB | ResNet32 | 5.1M | 19.45MB |
| MEMO | 3458 | 10.13MB | ResNet32 | 3.98M | 15.17MB |

Table 22: Configurations of different methods under CIFAR100 Base50 Inc10 setting.

| 16.45MB | $|\mathcal{E}|$ | $S(\mathcal{E})$ | Model Type | # Parameters | Model Size |
|---|---|---|---|---|---|
| Replay | 5017 | 14.7MB | ResNet32 | 0.46M | 1.75MB |
| iCaRL | 5017 | 14.7MB | ResNet32 | 0.46M | 1.75MB |
| WA | 5017 | 14.7MB | ResNet32 | 0.46M | 1.75MB |
| DER | 2000 | 5.86MB | ResNet32 | 2.78M | 10.61MB |
| MEMO | 2729 | 8.0MB | ResNet32 | 2.22M | 8.45MB |

Table 23: Configurations of different methods under ImageNet100 Base50 Inc5 setting.

| 756.1MB | $|\mathcal{E}|$ | $S(\mathcal{E})$ | Model Type | # Parameters | Model Size |
|---|---|---|---|---|---|
| Replay | 4970 | 713.47MB | ResNet18 | 11.17M | 42.63MB |
| iCaRL | 4970 | 713.47MB | ResNet18 | 11.17M | 42.63MB |
| WA | 4970 | 713.47MB | ResNet18 | 11.17M | 42.63MB |
| DER | 2000 | 287.1MB | ResNet18 | 122.94M | 468.99MB |
| MEMO | 2739 | 393.2MB | ResNet18 | 95.11M | 362.83MB |

Table 24: Configurations of different methods under ImageNet1000 Base0 Inc100 setting.

| 3297MB | $|\mathcal{E}|$ | $S(\mathcal{E})$ | Model Type | # Parameters | Model Size |
|---|---|---|---|---|---|
| Replay | 22672 | 3254.67MB | ResNet18 | 11.17M | 42.63MB |
| iCaRL | 22672 | 3254.67MB | ResNet18 | 11.17M | 42.63MB |
| WA | 22672 | 3254.67MB | ResNet18 | 11.17M | 42.63MB |
| DER | 20000 | 2871MB | ResNet18 | 111.76M | 426.35MB |
| MEMO | 20666 | 2967MB | ResNet18 | 86.72M | 330.81MB |

# B    VARIATIONS OF MEMO

In this section, we discuss the variations of MEMO , including the choice of specialized blocks and implementation with other backbones.

## B.1    HOW TO DEFINE THE SPECIALIZE AND GENERALIZE BLOCKS?

In the main paper, we observe the differences between shallow and deep layers in terms of gradient, MSE, and similarity. Hence, we decouple the network structure into deep and shallow layers and treat the deep layers as specialized blocks and the others as generalized ones. It is worth exploring how to decompose the model into specialized and generalized blocks. In this section, we conduct vast experiments to give the rule of thumb for defining the specialized and generalized blocks.

We first take ResNet32 as an example. There are three groups of residual blocks in ResNet32,[2] *i.e.*, residual blocks $1 \sim 5$ are encapsulated as group 1; residual blocks $6 \sim 10$ are encapsulated as group 2 and $11 \sim 15$ as group 3. We treat these groups as the minimal unit when decoupling the network.

Since there are only three groups of blocks in ResNet32, there are only two ways to decouple the network structure (cutoff after group 1 or 2). Therefore, we conduct experiments to compare the performance when treating the last group as the specialized block (as discussed in the main paper) and the last two groups as specialized blocks (denoted as MEMO-2). MEMO-2 extends two groups of

---

[2]These groups are denoted as 'layers' in the implementation (https://github.com/pytorch/vision/blob/main/torchvision/models/resnet.py), and each 'layer' may contain several residual blocks. We interchangeably use 'layers' and 'groups' in this paper, and both of them denote the combination of residual blocks.

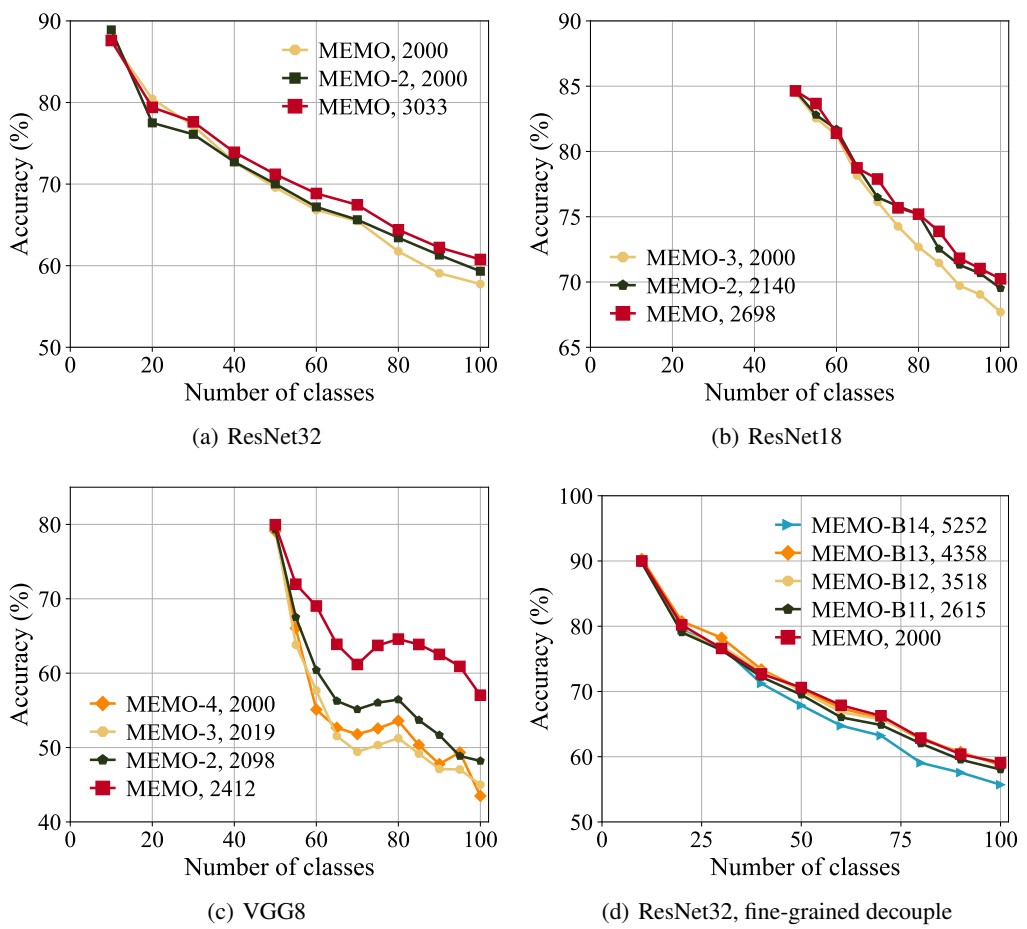

(a) ResNet32

(b) ResNet18

(c) VGG8

(d) ResNet32, fine-grained decouple

Figure 19: Experiments about the choice of specialized and generalized blocks. The digit after model name denotes the number of exemplars used during model training. **Choosing the last block groups as specialized blocks is more memory-efficient.**

residual blocks at a time, which consumes more memory budget than MEMO. We follow the same evaluation protocol as Section 5.1 in the main paper to compare under the CIFAR100 Base0 Inc10 setting, and report the results in Figure 19(a).

There are three lines in Figure 19(a), and the number of exemplars is shown in the legend. **It is obvious that** MEMO**-2 uses more memory size than** MEMO **with the same exemplar size**. Hence, we follow the description in the main paper and align the memory cost of MEMO to MEMO-2, and denote the method with aligned exemplars as MEMO**, 3033**. There are two main conclusions in this figure. Firstly, MEMO-2 has a little better performance than MEMO with the same exemplar size, which means extending more generalized blocks can improve the performance, although they are highly similar. But it should be noted that these methods are not fairly compared since MEMO-2 uses more model size than MEMO. Secondly, when aligning the memory cost of MEMO, 3033 to MEMO-2, we can infer that MEMO has better performance than MEMO-2, verifying that treating the last layer as specialized blocks is more memory-efficient. In other words, saving the generalized blocks per task is less efficient than saving exemplars of equal size.

We also explore the strategy with ResNet18 under the ImageNet100 Base50 Inc5 setting and report the results in Figure 19(b). Since there are four layers in ResNet18, we choose to decouple the network after the first, second, and third layers, resulting in MEMO-3, MEMO-2, and MEMO, respectively. Furthermore, we align the memory budget of different methods to MEMO-3 since it costs the largest budget. The ranking of different variations is the same as Figure 19(a), and we find treating the

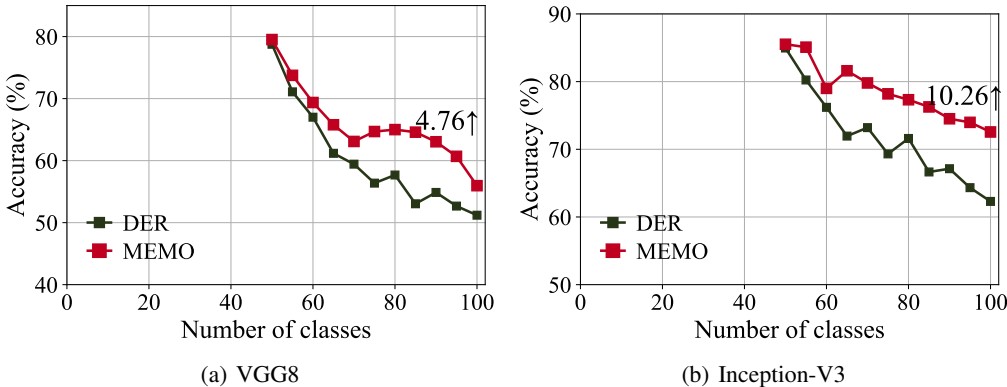

(a) VGG8                   (b) Inception-V3

Figure 20: Experiments when varying the network structure. We evaluate MEMO and DER with VGG8 and Inception-V3 on ImageNet100. MEMO **consistently outperforms DER with different network structures.**

last group of blocks as the generalized block is more memory-efficient, which consistently shows the best performance. We also verify this conclusion regarding other backbone structures, *i.e.*, VGGNet (Simonyan & Zisserman, 2014). There are five layers in it, and we follow the same protocol to conduct the experiments and show the results in Figure 19(c). Observing the results with different network structures and datasets, we empirically find that choosing the last layer as the specialized block is more memory-efficient.

Since the groups are combined with residual blocks, we can also decouple the network from the middle of the layers. Taking ResNet32 as an example, we can decouple the network after residual block 11, after residual block 12, etc. We also experiment following the former settings, and the results are shown in Figure 19(d). We denote the variation which decouples the network after the n-th residual block as MEMO-Bn. It implies that decoupling from the middle of the group is not a good choice. A probable reason is that the decoupling harms the inner characteristics of the residual groups, resulting in inferior performance. Hence, we do not conduct fine-grained model decoupling and treat the residual groups as the minimal unit in model decoupling.

To summarize, we suggest **treating the last residual group in the network as specialized blocks when decoupling the layers**, which is proven to be most memory-efficient among different network structures and datasets. This rule is consistent with the observations in Figure 3 that the last layer shifts more than others.

## B.2 MEMO WITH OTHER BACKBONES

As discussed in the main paper, our implementation is based on ResNet, but the concept of MEMO can be applied to any other deep network structure that relies on deep and shallow features. In this section, we conduct experiments with VGGNet (Simonyan & Zisserman, 2014) and Inception (Szegedy et al., 2016) on ImageNet100, and compare MEMO with DER with the same backbone. Results are shown in Figure 20. Other settings are the same as in the main paper.

We use VGG8 and Inception-V3 for implementation. VGG8 is a relatively small backbone, and we can exchange the extra model size of DER into 412 exemplars for MEMO. Inception V3 is much larger, and the additional number of exemplars is 2512 for Inception-V3. We can infer from these figures that MEMO shows consistent improvement over DER on these different network backbones, verifying that MEMO is a generalized protocol and can be applied to various kinds of CIL tasks with various network structures.

## C    Extra experimental evaluations

In this section, we give the extra experimental evaluations, including the last accuracy-memory curve, the gradient norm of all tasks, CKA visualization of different layers, multiple runs, and running time. We also conduct experiments on ImageNet to discuss which layer should be frozen in MEMO.

### C.1    Last Accuracy-Memory Curve

There are two commonly used performance measures for class-incremental learning. Denote the test accuracy after the $b$-th stage as $\mathcal{A}_b$, the average accuracy $\bar{\mathcal{A}} = \frac{1}{B} \sum_{i=1}^{B} \mathcal{A}_i$ represents model's average performance with streaming data. The last accuracy $\mathcal{A}_B$ denotes the performance after the last learning stage. These two accuracy measures are commonly used to measure the CIL performance in former works (Rebuffi et al., 2017; Zhao et al., 2020; Yu & Aizawa, 2019; Wu et al., 2019). We provide the performance-memory curve with average accuracy in the main paper and report the curve with the last accuracy in Figure 21.

We can infer from these figures that the observations in the main paper still hold, *e.g.*, we observe the intersection between exemplar-based methods and model-based methods in CIFAR100. We do not observe the intersection on ImageNet100, but the performances of DER and WA are relatively close at the start point. The main reason is that the performance of the 4-layer ConvNet with a million parameters is enough to obtain diverse feature representations (as discussed in Figure 13). We can infer from Figure 21(b) that the intersection will be observed given a smaller memory size. Besides, the rankings between methods are not changed between these methods, and MEMO outperforms others by a substantial margin in most cases.

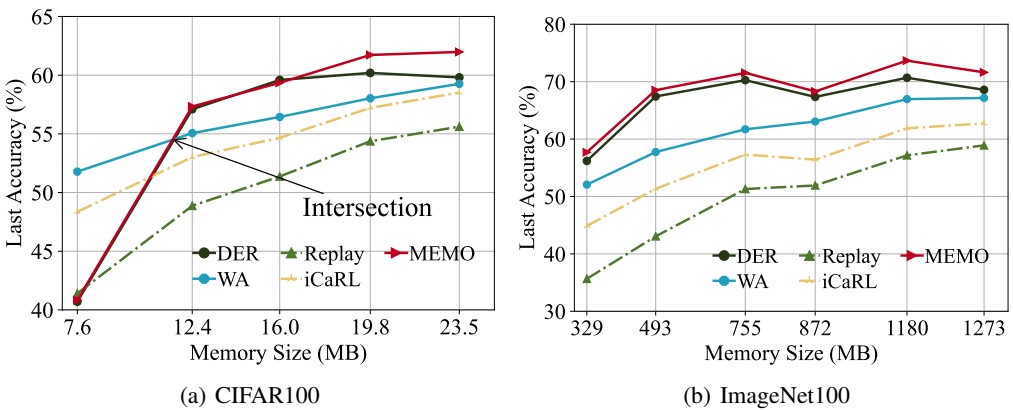

(a) CIFAR100                             (b) ImageNet100

Figure 21: Last accuracy-memory curve for CIFAR100 and ImageNet100. Other settings are the same as the main paper. **The order between these methods still holds.**

### C.2    Which Layer Should be Frozen?

In the main paper, we discuss the choice of fixing/unfixing the specialized and generalized blocks with empirical evaluations on CIFAR100. We report similar observations on ImageNet100 in Figure 22.

The main conclusions of these datasets are consistent with the former ones. Firstly, specialized blocks should be frozen. This conclusion is observed by comparing the results that $\bar{\phi}_s(\phi_g(\mathbf{x}))$ has better performance than $\phi_s(\phi_g(\mathbf{x}))$, and $\bar{\phi}_s(\bar{\phi}_g(\mathbf{x}))$ has better performance than $\phi_s(\bar{\phi}_g(\mathbf{x}))$. Secondly, by comparing the best strategy in the different settings, we can infer that fixing or not the generalized block depends on the number of classes in the first incremental task. It can be seen that $\bar{\phi}_s(\phi_g(\mathbf{x}))$ has the best performance with 10 base classes, while $\bar{\phi}_s(\bar{\phi}_g(\mathbf{x}))$ has the best performance with 50 base classes. The main reason behind these phenomena is the definition of a 'good' generalized block. When the base classes are large enough, training these classes can obtain diverse and transferable generalized representations. By contrast, if the base classes are small with few classes, training these

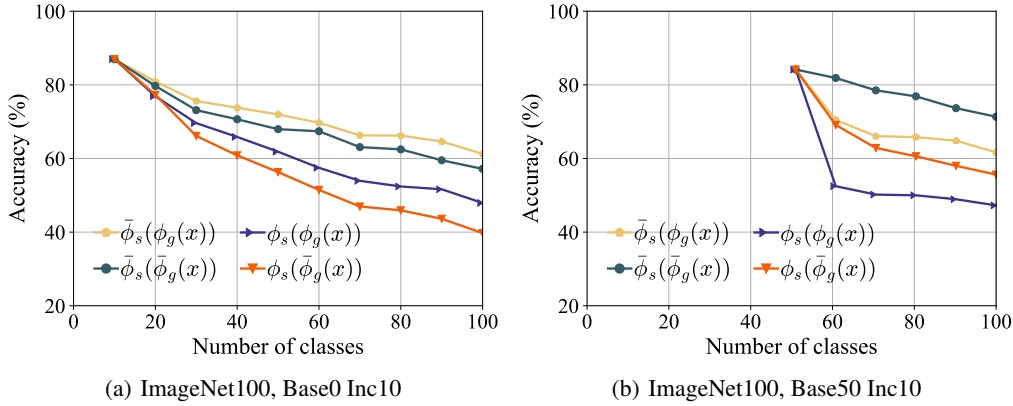

(a) ImageNet100, Base0 Inc10

(b) ImageNet100, Base50 Inc10

Figure 22: Experiments about specialized and generalized blocks. **Specialized blocks should be fixed, while fixing or not generalized blocks depends on the number of classes in the base stage.**

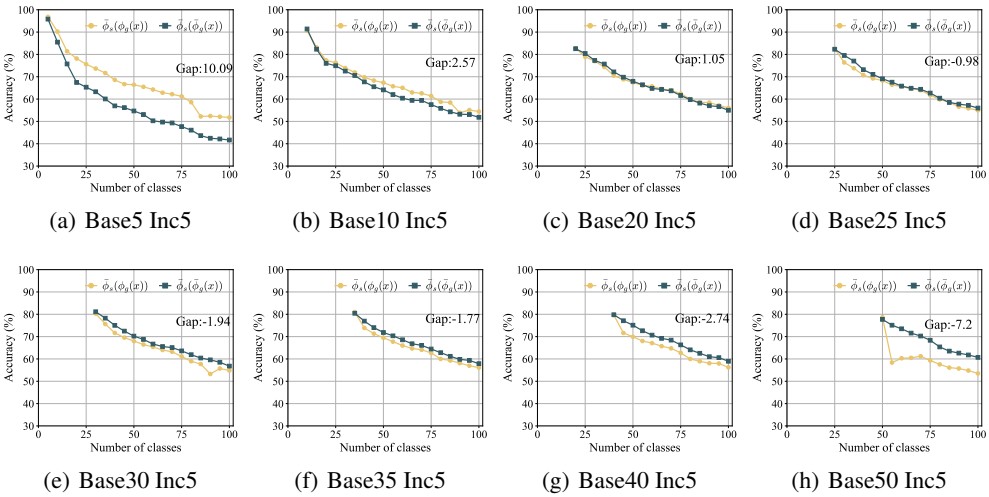

(a) Base5 Inc5

(b) Base10 Inc5

(c) Base20 Inc5

(d) Base25 Inc5

(e) Base30 Inc5

(f) Base35 Inc5

(g) Base40 Inc5

(h) Base50 Inc5

Figure 23: Comparison of $\bar{\phi}_s(\phi_g(\mathbf{x}))$ and $\bar{\phi}_s(\bar{\phi}_g(\mathbf{x}))$ given differnt base classes on CIFAR100 dataset. We annotate the gap of the last accuracy between $\bar{\phi}_s(\phi_g(\mathbf{x}))$ and $\bar{\phi}_s(\bar{\phi}_g(\mathbf{x}))$ at the end of each line. **Fewer base classes prefer trainable generalized blocks, while more base classes prefer frozen generalized blocks. The intersection among them emerges around 20 base classes.**

classes cannot obtain diverse and transferable generalized representations, and fixing the generalized block will harm the representation ability of the model and the final performance.

In real-world applications, we may not know the incremental learning implementations in advance. Hence, it requires the model to design a suitable strategy to automatically choose the frozen layers. In Figure 23, we show a preliminary experiment by changing the number of base classes in CIFAR100. Specifically, we vary the number of base classes among {5, 10, 20, 25, 30, 35, 40, 50} and show the incremental performance. In the former experiments, we already know that specialized blocks should be frozen. Hence, we only compare $\bar{\phi}_s(\phi_g(\mathbf{x}))$ and $\bar{\phi}_s(\bar{\phi}_g(\mathbf{x}))$ in these figures.

As we can infer from these figures, $\bar{\phi}_s(\phi_g(\mathbf{x}))$ (*i.e.*, do not freeze the generalized blocks) shows better performance when the base classes are limited, *e.g.*, fewer than 20. The gap between these strategies becomes smaller as the number of base classes increases. When the base class number reaches 20, the intersection among them emerges. After that, $\bar{\phi}_s(\bar{\phi}_g(\mathbf{x}))$ (*i.e.*, freezing the generalized blocks) shows better performance. Hence, we can treat 20 as an empirical threshold to define the learning protocol.

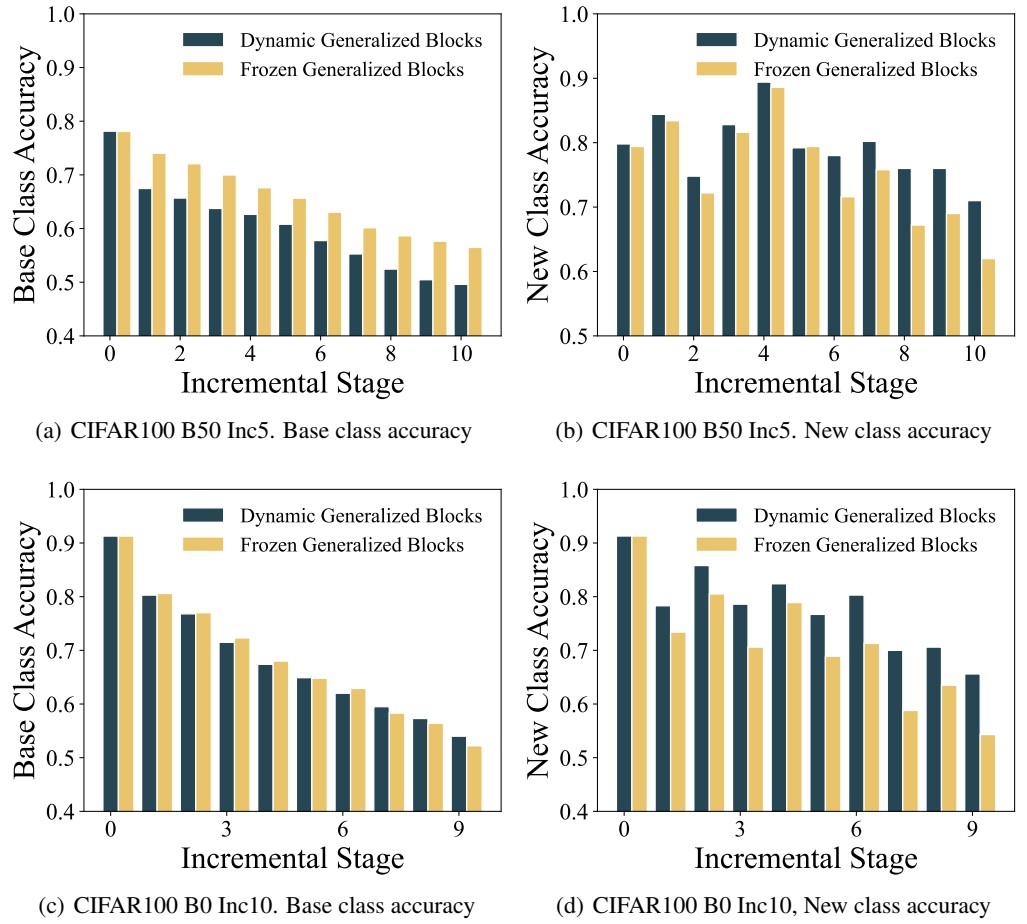

Figure 24: Base and new class accuracy of different strategies. **Freezing the generalized blocks helps to overcome catastrophic forgetting, while changing generalized blocks helps to adapt to new classes efficiently.**

It must be noted that these figures only correspond to a specific case of the CIFAR100 dataset, which may be different in other learning scenarios. Designing proper metrics to measure the generalizability of the shallow layers is interesting future work, and a possible solution is to utilize the hold-out or wild data for evaluation. On the other hand, it would be interesting to explore how to strengthen the generalization ability of shallow layers, *e.g.*, via meta-learning (Sun et al., 2019), self-supervised learning (Khosla et al., 2020) and deep metric-learning (Wang et al., 2017).

## C.3 INFLUENCE OF FREEZING GENERALIZED BLOCKS

In Section C.2, we empirically verify that the strategy to freeze or not the generalized blocks is related to the number of base classes. In this section, we explore the influence when freezing the generalized blocks. Specifically, we run the experiment under CIFAR100 B0 Inc10 and CIFAR100 B50 Inc5 setting and report the results in Figure 24.

In these figures, we separately record the accuracy of the 'Base' and 'New' classes along incremental stages. 'Base' classes denote the classes in the first incremental stage, *i.e.*, in $\mathcal{D}^0$, while 'New' classes represent the classes in the latest incremental stage, *i.e.*, in $\mathcal{D}^b$. As a result, the accuracy of base classes represents the ability to resist catastrophic forgetting, while the accuracy of new classes implies the ability of the model to adapt to new classes. These abilities are also known as 'stability-plasticity

dilemma' (Grossberg, 2012). It should be noted that incremental performance jointly considers the performance among all seen classes, and both abilities are essential in incremental learning.

As we can infer from these figures, freezing generalized blocks shows better performance on the base classes, which means it can better resist catastrophic forgetting. However, when it comes to new classes, dynamic generalized blocks show to have better performance. The reason is intuitive that freezing the generalized blocks restricts the model from adapting to new patterns of new classes. In other words, the general layers are trained with the current task, which is not generalizable enough for new tasks if the current class data is insufficient. Under such circumstances, freezing these layers harms the learning ability of the model to adapt to new classes. On the other hand, since the model keeps an exemplar set of old class instances, jointly optimizing the general layers with exemplars and the current dataset helps to rectify it, making it generalize to all classes and enhancing the model's ability. It must be noted that the joint learning process can resist forgetting by rehearsing former instances.

To summarize, CIL requires the model to perform well among all seen classes. Hence, if the benefits of learning new classes surpass the loss of old classes, we should enable the model to adapt to new classes and not freeze the generalized blocks. By contrast, if there are numerous base classes and forgetting them shall significantly damage the performance, freezing the generalized blocks is a better solution. These results are consistent with Section C.2.

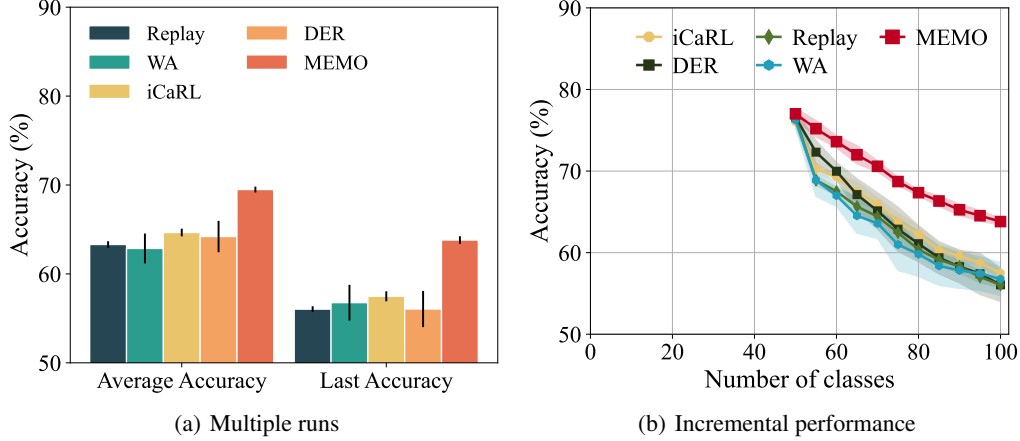

(a) Multiple runs             (b) Incremental performance

Figure 25: **Left**: The average and last accuracy of different methods. Error bars denote the standard deviation. **Right**: Average incremental performance among five runs. The standard deviation is shown in the shadow region. **The ranking of these methods is the same with different class orders.**

## C.4   INCREMENTAL LEARNING WITH MULTIPLE RUNS

A typical setting of CIL defines the comparison protocol (Rebuffi et al., 2017; Hou et al., 2019; Wu et al., 2019; Liu et al., 2020; Yu et al., 2020; Zhao et al., 2020; Yan et al., 2021; Douillard et al., 2021) to shuffle the class order with random seed 1993, and we follow this protocol to conduct the benchmark comparison in the main paper. In this section, we run the experiment multiple times with different random seeds and report the results in Figure 25(a). We choose random seeds from {10, 20, 30, 40, 50} and run the experiments five times with CIFAR100, Base50 Inc5. We also show the incremental performance of each method in Figure 25(b). We plot the average performance and standard deviation (shown in the shadow region) of each method among five runs. As we can infer from the figure, the results are consistent with the conclusions in the main paper, and the order of different classes remains the same with the change of random seeds.

## C.5   RUNNING TIME COMPARISON

Exemplar-based methods rely on revisiting former instances during new class learning, *i.e.*, the model optimizes the loss term over $\mathcal{D}^b \cup \mathcal{E}$ in every incremental stage. Hence, adding exemplar size will

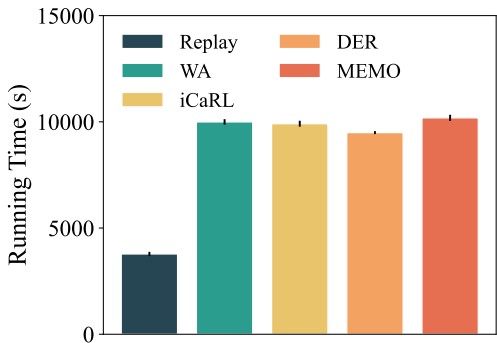

Figure 26: Running time comparison of different methods on CIFAR100. **The running time of** MEMO **is at the same scale as other methods.**

correspondingly increase the running time of these methods. On the other hand, model-based methods save the old backbones, which requires forwarding the same batch of instances multiple times with different backbones. These methods will increase the running time of class-incremental models, and we empirically analyze the running time of these methods on CIFAR100, Base0 Inc10 in this section. We report the running time in Figure 26.

As we can infer from the figure, simply replaying with exemplars consume the least running time, while it gets the worst performance among all methods. Adding the knowledge distillation can relieve catastrophic forgetting while substantially increasing the running time. On the other hand, expanding the models and saving old backbones obtains better performance. At the same time, it also increases the running time since a single instance should be forwarded by multiple backbones in the learning process. To summarize, we find that exemplar-based and model-based methods have the same scale running time. In other words, our MEMO achieves the best performance with the competitive running time, which is more efficient for developing CIL models in real-world applications.

## C.6 GRADIENT NORM OF ALL INCREMENTAL TASKS

In the main paper, we provide the gradient norm analysis for a single incremental task due to the page limit. We give the full gradients of all incremental tasks in this section, as shown in Figure 27. We can infer from these figures that the trend of gradient norm still holds for other incremental tasks. To be specific, the gradients of deeper layers are larger than shallow layers for all incremental tasks.

## C.7 CKA VISUALIZATION OF DIFFERENT LAYERS

In the main paper, we use CKA (Kornblith et al., 2019) to measure the similarity between different backbones learned during the incremental stages with ResNet32. We calculate the pair-wise feature similarities between the shallow layers (*i.e.*, after residual block 5) and deep layers (*i.e.*, after residual block 15) in the main paper. In this section, we provide the full CKA visualization of these residual blocks in Figure 28. As we can infer from these figures, the features of different backbones at the same depth yield different similarities. The features are highly similar for the shallow layers, *i.e.*, after residual block 5 and residual block 10. In contrast, the similarity is diverse for deeper layers, *i.e.*, after residual block 15. These results are consistent with the choice of generalized and specialized blocks discussed in the main paper and the empirical evaluations in Figure 19.

## C.8 COMPARISON WITH GAN-BASED METHODS

Generative models are capable of capturing the distribution of the data and generating instances. A typical line of work using GAN (Goodfellow et al., 2014) to memorize the distribution of former tasks and then replay them when learning new tasks (He et al., 2018; Shin et al., 2017a). Specifically, (Shin et al., 2017a) considers saving an extra GAN model as the generator and incrementally updates it when new data arrives. The classification model is optimized jointly with $\mathcal{D}^{new} \cup \mathcal{D}^{gen}$, where

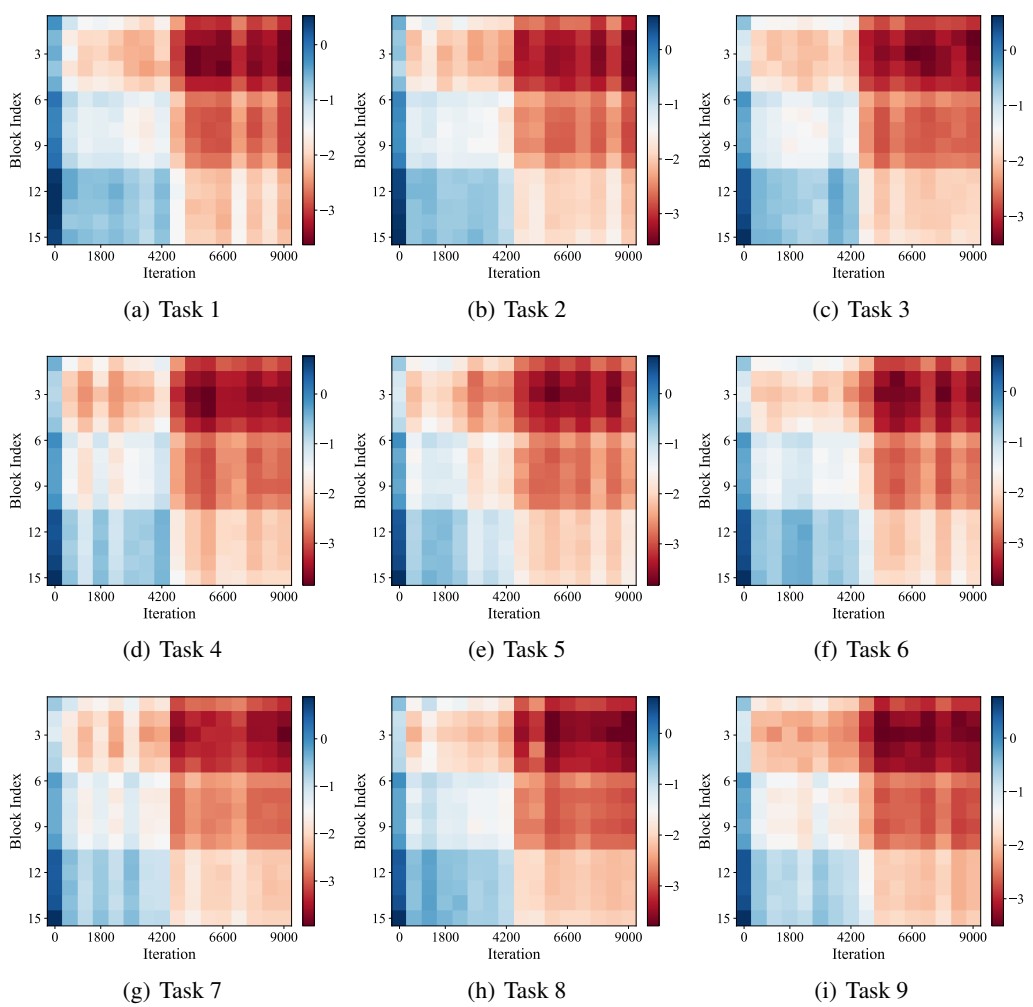

(a) Task 1      (b) Task 2      (c) Task 3

(d) Task 4      (e) Task 5      (f) Task 6

(g) Task 7      (h) Task 8      (i) Task 9

Figure 27: Gradient norm of different tasks. The numbers are reported in log scale. **Deeper layers have larger gradients while shallow layers have smaller ones.**

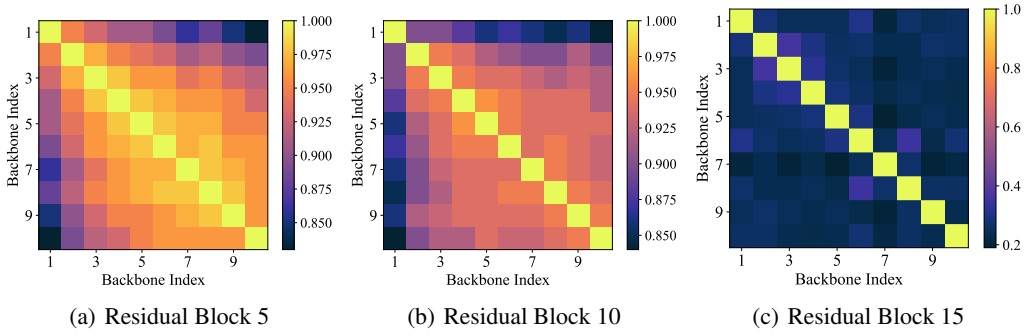

(a) Residual Block 5      (b) Residual Block 10      (c) Residual Block 15

Figure 28: CKA visualization of different layers. The larger block index indicates deeper layers. **The features extracted by deeper layers of different backbones are dissimilar, while that of shallow layers are similar.**

$\mathcal{D}^{new}$ stands for the incoming dataset and $\mathcal{D}^{gen}$ stands for the generated dataset from the former distribution. However, incrementally updating a single GAN model will also incur catastrophic

forgetting. Hence, (He et al., 2018) considers training a new GAN per incremental class to resist the forgetting phenomena in GAN updating. It requires saving multiple generative models in the memory, which costs more memory as the data stream evolves. Apart from saving GANs, (He et al., 2018) finds it useful to save exemplars from the former distribution, and optimizes the model with $\mathcal{D}^{new} \cup \mathcal{D}^{gen} \cup \mathcal{E}$.

In this section, we re-implement GR (Shin et al., 2017a) and ESGR (He et al., 2018) and compare them to our proposed MEMO with CIFAR100 dataset. We follow the implementations in the main paper to organize a 'Base 0 Inc 10' setting. ESGR requires saving multiple GANs, which costs a much higher memory budget, and we do not align the cost of it to the others. We follow the original paper to use WGAN (Arjovsky et al., 2017) as the generative model and implement it with four transposed convolutional layers. The optimization details (rounds, learning rate, optimizer) are set according to the original paper. We report the results in Table 25 and Figure 29.

We can infer from these results that training a GAN costs a large number of parameters, which is also hard to optimize for complex image inputs. Our proposed MEMO outperforms these methods by a substantial margin, even with a much fewer memory budget. As a result, we do not compare MEMO to these GAN-based methods in the main paper.

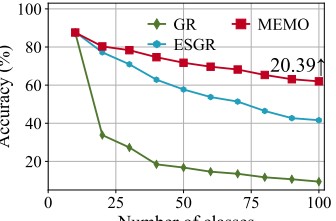

| Method | $|\mathcal{E}|$ | $S(\mathcal{E})$ | Model Type | # Parameters | Model Size |
|--------|------|----------|------------|--------------|------------|
| GR | 3658 | 10.71MB | ResNet32 + WGAN | 3.35M | 12.78MB |
| ESGR | 3300 | 9.66MB | ResNet32 + 100*WGAN | 166.39M | 634.75MB |
| MEMO | 3300 | 9.66MB | ResNet32 | 3.62M | 13.83MB |

Table 25: Implementation details when comparing to GAN-based methods.

Figure 29: CIL performance

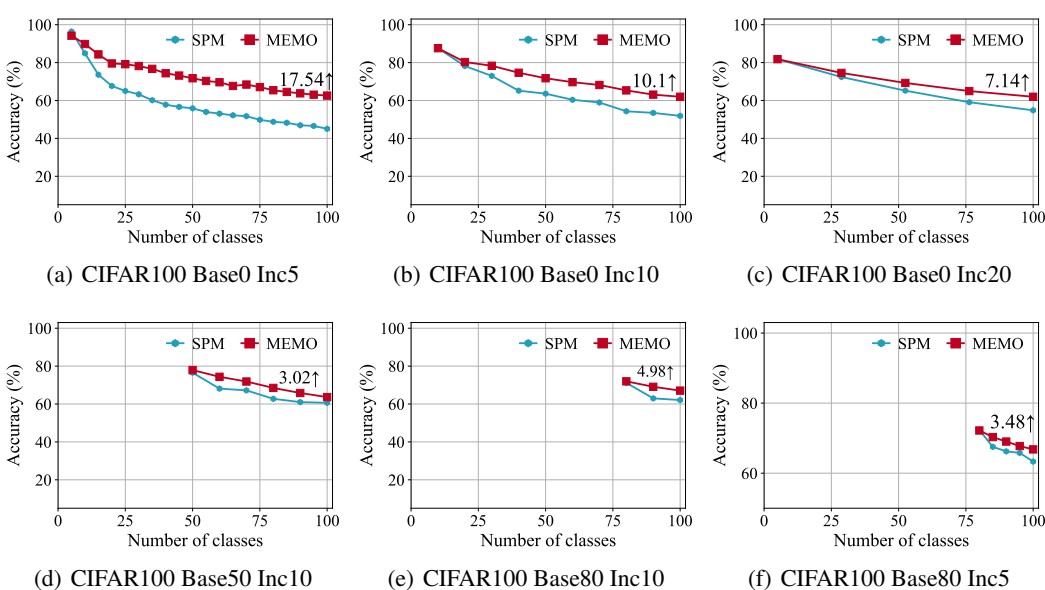

(a) CIFAR100 Base0 Inc5     (b) CIFAR100 Base0 Inc10     (c) CIFAR100 Base0 Inc20

(d) CIFAR100 Base50 Inc10     (e) CIFAR100 Base80 Inc10     (f) CIFAR100 Base80 Inc5

Figure 30: Experiments when comparing our proposed method to SPM. MEMO **consistently outperforms SPM with different benchmark settings.**

### C.9 COMPARISON WITH MULTI-BRANCH MODEL

A recent work SPM (Wu et al., 2022) also considers the multi-branch model for class-incremental learning. It should be noted that SPM is designed for CIL with vast base classes, *e.g.*, with 500 or 800 classes in the first stage. However, there are not so many base classes in the benchmark CIL setting,

and we re-implement SPM under the benchmark setting for comparison. In the implementation, we train a model from scratch with the base dataset for incremental learning.

Since SPM also decouples the backbone network into two parts, we follow the implementation in the original paper (Wu et al., 2022) to decouple the representation, *i.e.*, before the last convolutional block. Hence, the way of expansion in SPM is the same as MEMO, making the model size almost the same (the only difference lies in the fully connected layers, which is negligible).

Following the comparison protocol defined in the main paper, we can compare these methods fairly with the same exemplar size. Since the model size and memory size of SPM and MEMO is equal, the comparison is fair for them. The results are reported in Figure 30. We can infer from these figures that MEMO consistently outperforms SPM on these benchmark settings. The comparison to the contemporaneous multi-branch method verifies the effectiveness of our proposed method.

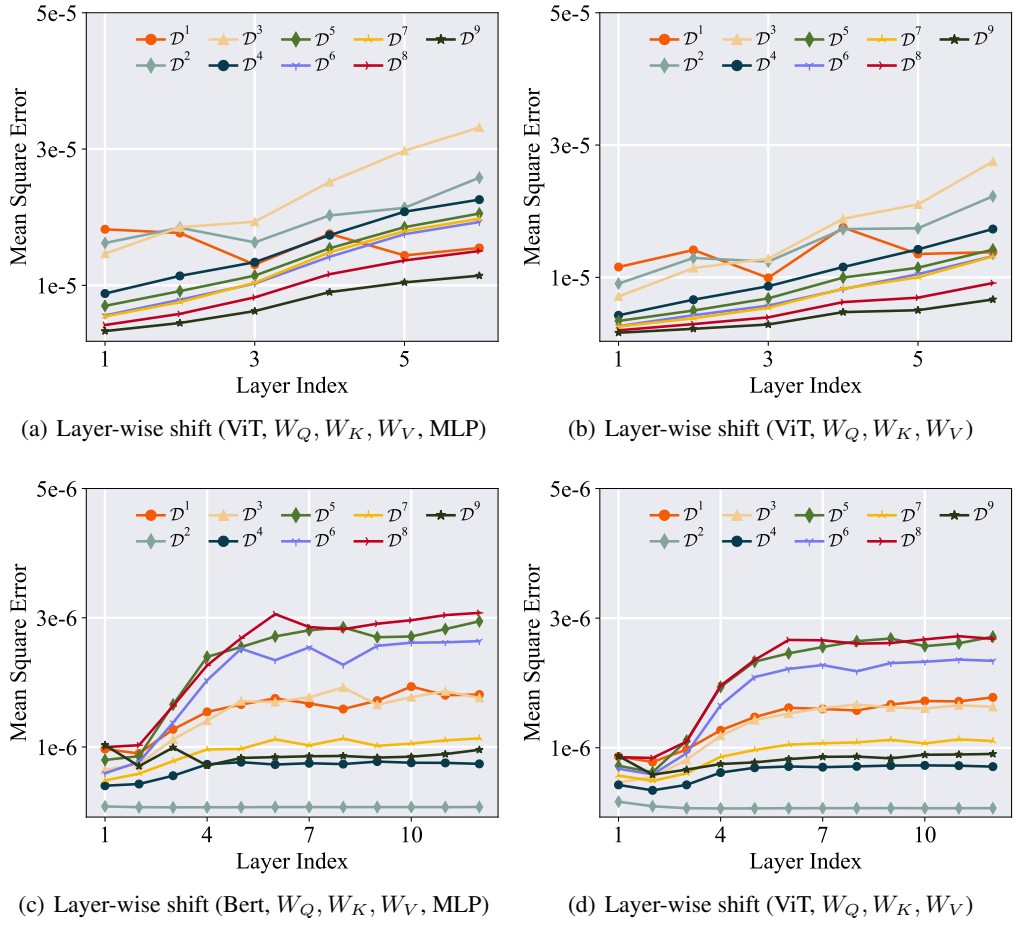

(a) Layer-wise shift (ViT, $W_Q, W_K, W_V$, MLP)

(b) Layer-wise shift (ViT, $W_Q, W_K, W_V$)

(c) Layer-wise shift (Bert, $W_Q, W_K, W_V$, MLP)

(d) Layer-wise shift (ViT, $W_Q, W_K, W_V$)

Figure 31: Layer-wise shift of ViT and Bert. The observations in residual networks still hold for these network structures and NLP datasets.

## C.10 DIFFERENT BACKBONES AND DATASETS

In the main paper, we explore the network behavior with ResNet (He et al., 2015) on image datasets, and find shallow layers stay unchanged while deep layers change more in class-incremental learning. In this section, we verify this conclusion with different network structures and different incremental datasets.

Specifically, we first explore the behaviors of Vision Transformer (Dosovitskiy et al., 2020) in incremental tasks. We directly run the experiments on CIFAR100 Base0 Inc10 with a 6-layer ViT.

Following the experiments in the main paper, we trace the weights in each Transformer encoder (including the weights of multi-head attention and MLP) and show the trends in Figure 31(a), 31(b). In Figure 31(a), we can observe that the layer-wise shift of deeper layers is more obvious than shallow ones. We also trace the shift of multi-head attention blocks, *i.e.*, $W_Q, W_K, W_V$ in Figure 31(b), and find the trend is consistent.

Additionally, we explore the behaviors of the network in NLP tasks. We follow (Ke et al., 2021) to use Bert (Devlin et al., 2018) for ASC dataset (Ke et al., 2021) (the aspect sentiment classification dataset containing 19 products), which is a benchmark NLP task. We keep the other settings the same and trace the shift of Transformer layers in Bert and draw the trends in Figure 31(c), 31(d). These results show the same trend as ViT.

As we can infer from these figures, the layer-wise change shows that shallow layers change less while deep layers change more, sharing the same trend as in residual networks. Hence, we can summarize that in various settings, shallow layers stay unchanged while deep layers change more in class-incremental learning.

## D IMPLEMENTATIONS

### D.1 DISCUSSIONS ABOUT RELATED WORK

Class-incremental learning is now a hot topic in the machine learning field, where new methodologies emerge frequently. We discuss CIL methods by dividing them into two groups, *e.g.*, exemplar-based and model-based. These groups either save exemplars or models to boost their performance.

However, there are other methods that do not fall into these groups, *e.g.*, (Kirkpatrick et al., 2017; Li & Hoiem, 2017; Jin et al., 2021; PourKeshavarzi et al., 2022). Since they are designed to conduct incremental learning without an extra memory budget, these methods often perform inferior to those discussed in the main paper, even equipped with additional exemplars. On the other hand, there are methods that address memory-efficiency from other aspects. For example, (Iscen et al., 2020) addresses memory-efficient CIL by saving the embeddings instead of raw images, (Zhao et al., 2021c) suggests saving low-fidelity exemplars, and (He et al., 2018; Shin et al., 2017a) address the problem by generating exemplars with generative models. It should be noted that saving embeddings will suffer the embedding drift phenomenon (Yu et al., 2020) and requires an extra finetuning process to fix such drift. The bias of the embedding adaptation process will accumulate, which works poorly in incremental learning with multiple stages. We re-implement (Iscen et al., 2020) and find it shows inferior performance than saving exemplars. Generating exemplars with generative models will consume the memory budget to save generative models, which also suffer catastrophic forgetting (Cong et al., 2020) and fail for large-scale images (Iscen et al., 2020) (See Section C.8). There are other methods designing multi-branch network structures, *e.g.*, AANets (Liu et al., 2021a), PNN (Rusu et al., 2016) and SPM (Wu et al., 2022). The extra branch is designed to re-scale the network outputs (Liu et al., 2021a) or get multi-head predictions (Wu et al., 2022), which requires an extra tuning process or routing design. By contrast, our proposed method does not require the extra model tuning and shows stronger performance (See Section C.9). As a result, we mainly concentrate on the discussions about typical CIL methods that rely on extra memory from the efficiency and model complexity perspective. Below are the introductions to compared methods in this paper.

- Replay (Chaudhry et al., 2019) is an exemplar-based baseline that simply optimizes the cross-entropy loss with $\mathcal{D}^b \cup \mathcal{E}$ in every incremental stage;
- iCaRL (Rebuffi et al., 2017) builds knowledge distillation (Zhou et al., 2003; Zhou & Jiang, 2004; Hinton et al., 2015) regularization term to regularize former classes from being forgotten. The loss (cross-entropy and knowledge distillation) is optimized with $\mathcal{D}^b \cup \mathcal{E}$ in every incremental stage;
- WA (Zhao et al., 2020) extends iCaRL with weight aligning, which normalizes the linear layers to reduce the negative bias;
- DER (Yan et al., 2021) is a model-based method that saves backbones to resist catastrophic forgetting. Apart from the loss term discussed in the main paper, it also introduces an auxiliary loss to encourage diverse representations and a sparse loss to conduct network pruning.

In this paper, we implement DER with two modifications to the original implementation. Firstly, the original DER utilizes different backbone networks than other methods, *e.g.*, modified ResNet18 for CIFAR100, while other methods utilize ResNet32 as the benchmark backbone. In this paper, we report the benchmark comparison when using the same backbones for DER and other exemplar-based methods (in Section 5.1 of the main paper). Secondly, DER claims to use a pruning algorithm to reduce the parameter number. But it shows that the pruning has little effect on the total parameters and harms the final performance. On the other hand, the pruning code is not open-sourced yet.[3] As a result, we re-implement DER without the pruning loss to report the results in this paper.

Recently, SPM (Wu et al., 2022) finds that fixing the generalized block is effective for class-incremental learning with vast base classes and proposes a multi-branch method for CIL with strong pre-trained models. *The findings in SPM only correspond to a tiny portion of ours*, and we list the differences below:

- **Different Settings:** SPM concentrates on the incremental learning scenario with a strong pre-trained model, which requires 500 or 800 base classes in their setting. However, we are focusing on the typical class-incremental learning setting, where the model should be trained from scratch with few classes (say, 5 or 10). Our analysis in Section 4.3 is also irrelevant to the strong pre-trained models.

- **Different Observations:** The observation in SPM is highly driven by their setting, i.e., they focus on how to design incremental models with vast base classes and find freezing the generalized block can facilitate model learning. However, the findings in SPM are only a tiny portion of our conclusions in Figure 5. Apart from the aforementioned findings, we also summarize that if the base classes are insufficient, fixing the generalized block will lead to inferior performance. Besides, we also find that the specialized blocks of former tasks should be frozen in all cases, which is also not discussed in SPM.

- **Different Network Behaviors:** It should be noted that none of the conclusions in Section 4.3 can be found in SPM, where we analyze from the aspect of network behaviors (gradient norm, shift between blocks, and CKA between backbones). As a result, the conclusions in SPM only correspond to a tiny portion of ours and should not weaken our novelty.

- **Different Methodologies:** Although our MEMO and SPM both adopt the multi-branch structure; the core difference is that SPM adds a new specialized block and fully-connected layer for each new task. It requires calibration between old and new classes, while ours will not face this problem since we update a larger fully-connected layer as Eq. 2 after each incremental stage. Besides, facilitated by the simple and effective training protocol, our method does not need any hyper-parameter to control the trade-off between loss terms.

We give the comparison of our proposed method to SPM in Section C.9, where we verify that our proposed method is more suitable for the benchmark class-incremental learning scenario.

## D.2 Implementation Details of Memo

Similar to DER, there are two loss terms in MEMO. Apart from the cross-entropy loss discussed in the main paper, the auxiliary loss aims to differentiate between old and new classes. Denote the $b$-th incremental stage dataset $\mathcal{D}^b$ contains $|Y_b|$ classes, we create an extra classifier $W_A \in \mathbb{R}^{d \times |Y_b|+1}$. The auxiliary loss is represented by:

$$\mathcal{L}_A(\mathbf{x}, \hat{y}) = \sum_{k=1}^{|Y_b|+1} -\mathbb{I}(\hat{y} = k) \log \mathcal{S}_k(W_A^\top \phi_b(\mathbf{x})), \tag{4}$$

where $\phi_b$ is the $b$-th embedding created for $\mathcal{D}^b$, $\hat{y}$ reassigns the ground-truth label into $|Y_b| + 1$ classes, and treat $y \notin Y_b$ as class $|Y_b| + 1$. Eq. 4 helps to acquire diverse feature representations for each embedding backbone, and the auxiliary classifier $W_A$ is dropped after each task training. The final optimization of MEMO combines the cross-entropy loss discussed in the main paper and the auxiliary loss in Eq. 4. We also use weight normalization to eliminate the bias in the classifier layer.

---

[3]https://github.com/Rhyssiyan/DER-ClassIL.pytorch

### D.3 EXEMPLAR SELECTION

As discussed in the main paper, we follow (Rebuffi et al., 2017) to select the exemplars in the exemplar set $\mathcal{E}$ with the herding algorithm (Welling, 2009). It aims to select the most representative instances per class by the distance to the class center. Denote the current embedding as $\phi(\cdot)$. Given the instance set $X = \{\mathbf{x}_1, \mathbf{x}_2, \cdots, \mathbf{x}_n\}$ from class $y$, the target is to select $m$ representative instances from $X$. We first calculate the class mean via: $\mu \leftarrow \frac{1}{n} \sum_{i=1}^{n} \phi(\mathbf{x}_i)$. We then calculate and rank the distance of each instance to the class center $\|\mu - \phi(\mathbf{x}_i)\|$ in ascending order. The exemplar set $\mathcal{E}$ is the collection of the top-$m$ instances with the least distance.

### D.4 MODEL COMPRESSION

This paper mainly discusses *how to allocate the memory budget for class-incremental models*, *i.e.*, investigating a memory-efficient way to allocate the model and exemplar given a specific memory budget. We do not aim to design extra model compression algorithms or post tuning methods to reduce the total parameters. By contrast, we concentrate on *which part of the model should be saved/dropped*. Given the fact that there are many methods to conduct model compression (Deng et al., 2020), *e.g.*, knowledge distillation (Wang et al., 2022), pruning (Yan et al., 2021), low-rank factorization (Sainath et al., 2013) and quantization (Yang et al., 2019), we believe they can be combined with our method orthogonally as interesting future work.

### D.5 BROADER IMPACT

In this work, we study the class-incremental learning problem, a fundamental problem in machine learning. Specifically, we first address the fair comparison among different methods by aligning the memory budget and propose several performance measures for a holistic evaluation. We then observe the differences between different layers in the class-incremental learning process and propose a simple yet effective baseline method to efficiently organize the memory budget. Our work will give instructions for applications with difficulties managing the memory size for CIL models. At the same time, there is still much room for exploration in this work. We hope our work can inspire more discussions about class-incremental learning in real-world applications and drive more research to build practical and memory-efficient CIL models.

Meanwhile, we are aware that the abuse of this technology can pose ethical issues. In particular, we note that people expect that learning systems will not save any personal information for future rehearsal. While there are risks with this kind of AI research, we believe that developing and demonstrating such techniques is essential for understanding valuable and potentially troubling applications of the technology. We hope to stimulate discussion about best practices and controls on these methods around responsible technology uses.

