# OpenReview forum: "A Model or 603 Exemplars: Towards Memory-Efficient Class-Incremental Learning"
_ICLR.cc/2023/Conference — ICLR 2023 notable top 25%_

### Official Review · Reviewer_FkFJ · 2022-10-23

**Confidence:** 4
**Correctness:** 3
**Technical Novelty And Significance:** 2
**Empirical Novelty And Significance:** 3
**Recommendation:** 6

**Clarity, Quality, Novelty And Reproducibility:**

The authors present their motivations clearly, however the contributions are not completely new, they are a small step to what was already known.

I leave a few questions to the authors regarding the experiments.I think the experiments are good, but they are not capable of answering all the questions raised as motivations by the authors.
For example, one of the questions that I did not see answered is where to make the cut between global and specific weights. I think this is a fundamental point that the authors do not dispute. I understand the space limitations, but it is a very important point.

**Strength And Weaknesses:**

S:
- Based on current works, the community agrees with the authors that there is a need to improve how we compare methods. A suitable example of this is the framework proposed in [1].
- The idea of reducing the number of layers saved in the "memory buffer" to save more examples or using better models is good. The authors' experiments that show that these layers store general information and take advantage of that, are intriguing.

W:
- The CL definition is unclear, making it unclear whether or not the authors employ task-ids during inference. If they use the same definition as Yan et al., 2021. One can assume they do not use the task id. However, it would be helpful to make it clear.
- "… Since a single backbone can depict a limited number of features, learning and overwriting new features will undoubtedly trigger the forgetting of old ones …" I would be careful with these kinds of statements. It may seem accurate at first, but we do not know how many concepts can be stored in each feature. For example, the model rarely used the entire weights based on the Lottery Tickets Hypothesis. Therefore, treating this as a limitation without giving further reasons is complex.
- In implementation details, it would be helpful to know how many times the experiments were run. Running the experiments only once is not good practice in CL, since the variance between different initializations and task orders can be significant.
- The motivation of Fig1 is straightforward but may be out of place. Several things are not explained until Sec4 (for example, Base50, Inc5, or MEMO). The explanation in the text could be improved or Fig1 could be left in Sec4.
- In sec4.2. Since the authors advocate a fair comparison, I do not think it is right to change the base model. Each model has its characteristics that can harm or benefit some methods or scenarios. The results in Fig2 validate this concern.
- Sec4.3. The experiments are interesting, but as the authors mention, it is something that is known. It may be possible to reduce this section and add experiments about how to properly divide general and specific layers. There is no mention in the paper of how this division is made.
- Sec5.2. If models are in a similar memory context, why not just compare ACC? The motivation for using AUC is not straightforward.

Q:
- Understand that this paper and [2] are different. In your opinion, what is the motivational difference between them?
- Could give an intuition for why not freezing the general layers improves the acc? If there is general knowledge in those layers, why change it? How much can we modify those layers and still prevent forgetting?

[1] Mundt, Martin, et al. "CLEVA-Compass: A Continual Learning EValuation Assessment Compass to Promote Research Transparency and Comparability." arXiv preprint arXiv:2110.03331 (2021).
[2] Ostapenko, Oleksiy, et al. "Continual learning via local module composition." Advances in Neural Information Processing Systems 34 (2021): 30298-30312.

**Summary Of The Paper:**

It is known that methods that store examples in memory have positive results for the Continual Learning problem, specifically in Class Incremental Learning. Instead of saving examples from previous tasks, some works recommended storing model checkpoints to retain the information. The authors of this paper point out that unfair comparisons are often performed since the model buffer is not always considered part of the saved memory. Because of this, the authors propose to compare the methods by adding both memories (exemplar and model), revealing that weight-saving methods tend to suffer when memory is limited. With these results, the authors propose MEMO, a more efficient method to store checkpoints. MEMO takes advantage of the fact that the first tends to have generic information, keeping those shared across tasks and only adding later layers to incorporate specific knowledge. The newly freed memory can then be used to save more examples of previous tasks. MEMO achieves comparable results and sometimes outperforms current methods in different scenarios.

**Summary Of The Review:**

Despite having limited contributions in my opinion, the ideas presented are good and may be interesting for the community.

However, the authors leave many questions raised that, if answered, could significantly improve the work. In my opinion, this research work is being developed and has not yet reached its maximum potential.

---

> ### Author Response · Authors · 2022-11-15
> **Response to Reviewer FkFJ (4/4)**
>
> **Q9** Could give an intuition for why not freezing the general layers improves the acc? If there is general knowledge in those layers, why change it? How much can we modify those layers and still prevent forgetting?
>
> **A9** We thank the reviewer for the suggestion. **In this revision, we have supplied the experimental results in Section C.3, illustrating the effect of freezing or changing general layers.** Specifically, we separately record the accuracy of the “Base” and “New” classes along incremental stages. “Base” classes denote the classes in the first incremental stage, i.e., in $\mathcal{D}^0$, while “New” classes represent the classes in the latest incremental stage, i.e., in $\mathcal{D}^b$. As a result, the accuracy of base classes represents the ability to resist catastrophic forgetting, while the accuracy of new classes implies the ability of the model to adapt to new classes. These abilities are also known as the “stability-plasticity dilemma.” It should be noted that incremental performance jointly considers the performance among all seen classes, and both abilities are essential in incremental learning.
>
> As we can infer from Figure 24, freezing generalized blocks shows better performance on the base classes, which means it can better resist catastrophic forgetting. However, when it comes to new classes, dynamic generalized blocks show to have better performance. The reason is intuitive that freezing the generalized blocks restricts the model from adapting to new patterns of new classes. In other words, the general layers are trained with the current task, which is not generalizable enough for new tasks if the current class data is insufficient. Under such circumstances, freezing these layers harms the learning ability of the model to adapt to new classes. On the other hand, since the model keeps an exemplar set of old class instances, jointly optimizing the general layers with exemplars and the current dataset helps to rectify it, making it generalize to all classes and enhancing the model’s ability. It must be noted that the joint learning process can resist forgetting by rehearsing former instances.
>
> To summarize, CIL requires the model to perform well among all seen classes. Hence, if the benefits of learning new classes surpass the loss of old classes, we should enable the model to adapt to new classes and not freeze the generalized blocks. By contrast, if there are numerous base classes and forgetting them shall significantly damage the performance, freezing the generalized blocks is a better solution.
>
>
> [1] Lifelong Learning with Dynamically Expandable Networks. ICLR18
>
> [2] iCaRL: Incremental Classifier and Representation Learning. CVPR17
>
> [3] Learning a Unified Classifier Incrementally via Rebalancing. CVPR19
>
> [4] Large Scale Incremental Learning. CVPR19
>
> [5] Mnemonics Training: Multi-Class Incremental Learning without Forgetting. CVPR20
>
> [6] Semantic Drift Compensation for Class-Incremental Learning. CVPR20
>
> [7] Maintaining Discrimination and Fairness in Class Incremental Learning. CVPR20
>
> [8] SS-IL: Separated Softmax for Incremental Learning. ICCV21
>
> [9] DER: Dynamically Expandable Representation for Class Incremental Learning. CVPR21
>
> [10] Prototype Augmentation and Self-Supervision for Incremental Learning. CVPR21
>
> [11] DyTox: Transformers for Continual Learning with DYnamic TOken eXpansion. CVPR22
>
> [12] How transferable are features in deep neural networks? NIPS14

---

> > ### Comment · Reviewer_FkFJ · 2022-11-16
> > **Thanks for the improvement of the document**
> >
> > I appreciate the detailed answers, the document updates, and the extra experiments performed. As a result of the high-quality answers, I felt it necessary to elaborate on my comments.
> >
> > Regarding **A2**. I disagree with this statement. I do not think it is valid to assert that the model's capabilities are full, and that this is why it is forgotten. The Lottery Tickets Hypothesis shows that it is possible to compress the information to represent a whole dataset in a subgroup of weights. This shows that the model contains redundant information in its weights. However, optimization is not encouraged in Continual Learning. If you apply fine-tuning, all the weights will change, not because the model is saturated, but because it is learning a new task. Nowhere in the optimization process is it intended to compress or save information. The overwriting of features is the problem, but not because of the limited number of features that can be saved. Nevertheless, this discussion is irrelevant to the work presented, so it is irrelevant to the final score. It was just a comment.
> >
> > Regarding **A5**. I agree with the authors: a good class-incremental learning algorithm should handle different budget restrictions. In the paper, experiments were performed with a defined budget, and I agree with the authors that this must be done. However, my concern is that: changing the architecture changes the results, regardless of whether the method and budget remain fixed. However, I understand that there are trade-offs and limits to the different experiments that can be done. Given the contributions and motivations, the experiments carried out are adequate.
> >
> > Concerning **A6**. I am not sure about the first point: Most papers that work with pre-trained models in continual learning (if we talk about general knowledge), can arrive at this conclusion in one way or another. However, the analysis added in Section B.1 helps address my initial concern.
> >
> > About **A7**. I appreciate the more detailed explanation. With the new explanation, I better understand the motivation for the metrics and agree that it can be beneficial to validate methods holistically.
> >
> > **A9**. If I understand correctly, the general layers are not that general at the beginning of the training process because they need to add information about new tasks, which makes sense since we have limited classes and knowledge. However, memory is enough to mitigate forgetting in these general layers. On the other hand, we need to train specific layers for each task because memory is not enough to mitigate forgetting in these layers.
> >
> > In summary, my main concerns were addressed in A6 and A7. For this reason, I am improving my score.

---

> > > ### Author Response · Authors · 2022-11-16
> > > **Many thanks!**
> > >
> > > Thank you for revising your score. We are happy we managed to address your concern.
> > >
> > > Regarding **A2**, we agree with the reviewer exploring the Lottery Tickets Hypothesis in class-incremental learning is interesting. For example, it would be meaningful to freeze the weights of the subnetwork and tune the rest parameters to evaluate the capacity usage. We will explore it in future work.
> > >
> > > Regarding **A9**, you are right, and the visualizations in Figure 7 also explain such phenomena.

---

> ### Author Response · Authors · 2022-11-15
> **Response to Reviewer FkFJ (3/4)**
>
> **Q7** Sec5.2. If models are in a similar memory context, why not just compare ACC? The motivation for using AUC is not straightforward.
>
> **A7** We thank the reviewer for the suggestion. As discussed in the main paper, a good class-incremental learning algorithm should work robustly given **any** memory budget. However, we can infer from Figure 1 that some methods work better with a small budget (e.g., iCaRL), while some prefer a large budget (e.g., DER). Hence, AUC is a **budget-agnostic** metric to **holistically** measure the performance given any budget. For example, if we are going to compare DER and iCaRL, we find iCaRL has better ACC when the memory size is 7.4MB, while DER works better when the memory size is 23.5MB. By contrast, AUC does not rely on the specific budget, and we can tell from Table 1 that DER has better AUC, which means it has better expandability.  In other words, ACC can only measure the performance given a specific X- coordinate, while AUC measures the area under the incremental performance curve holistically.
>
> **Q8** Understand that this paper and (Ostapenko et al., 2021) are different. In your opinion, what is the motivational difference between them?
>
> **A8** We thank the reviewer for the helpful related work. LMC (Ostapenko et al., 2021) aims to design dynamic modules in the network for feature transformation. These dynamic modules are specifically designed to generalize to related but unseen tasks. The model will expand dynamic modules when needed and combine different modules for the final prediction. However, since there may exist multiple modules in the network, LMC requires learning the pathway (or, say, dynamic routing) in the complex network to decide which module to use. However, the pathway is constant and simple in our proposed MEMO, and we do not rely on specific network structures to learn the pathway (e.g., invertible neural network or autoencoder).
>
> To summarize, LMC aims to enable different pathways in the incremental model and designs dynamic modules to achieve this goal. By contrast, the motivation of our MEMO is to extend the representation ability of the network with the least extra parameters. **In this revision, we have cited and discussed (Ostapenko et al., 2021) in the related work.**

---

> ### Author Response · Authors · 2022-11-15
> **Response to Reviewer FkFJ (2/4)**
>
> **Q5** In sec4.2. Since the authors advocate a fair comparison, I do not think it is right to change the base model. Each model has its characteristics that can harm or benefit some methods or scenarios. The results in Fig2 validate this concern.
>
> **A5** We thank the reviewer for the suggestion. In fact, a good class-incremental learning algorithm should handle different budget restrictions. For example, an algorithm should be able to learn with high-performance computers or with edge devices (e.g., smartphones), and both scenarios are essential in real-world applications. Former research concentrates on the adequate budget, where we can keep the large model in memory. However, when facing with limited budget, the algorithm should still work robustly. Hence, we aim for “fair comparison” among different methods and argue that different algorithms should be compared given the same memory budget.
>
> However, since some methods (i.e., DER and MEMO) require saving extra model components, directly comparing them to other methods with the same backbone will trigger the unfair comparison with different budgets. Hence, in Figure 1, we mainly use the same series model from ResNet (e.g., ResNet14/20/26) to reduce the burden of model size and align the total budget with other methods.
>
> **It should be noted that we also conduct experiments that do not change the network structure, i.e., comparing all the methods with the same backbone and budget in Section 5.1.** Please refer to Section 5.1 and A.3 for detailed evaluations, where MEMO still outperforms other methods by a substantial margin.
>
> **Q6** Sec4.3. The experiments are interesting, but as the authors mention, it is something that is known. It may be possible to reduce this section and add experiments about how to properly divide general and specific layers. There is no mention in the paper of how this division is made.
>
> **A6** We thank the reviewer for the suggestion. We correspond from two aspects.
>
> Firstly, the experiments in Section 4.3 have not been explored in other works. In other words, former works analyze the network behavior under static environments, while we are the first to explore them in incremental scenarios. For example, [12] finds that shallow layers are more transferable for downstream tasks. However, “being transferable” does not mean being unique, and it is unclear if they will stay unchanged when tuning them sequentially in the data stream. In this work, we find that shallow layers tend to remain static during incremental learning. In contrast, deep layers change severely from task to task, triggering the catastrophic forgetting of former tasks. **Hence, these results have yet to be discovered for the current incremental learning community.**
>
> Secondly, **we have suggested using the last layer as specialized blocks in the submitted version.** **In this revision, we have supplied experiments about network decoupling on various network structures, including ResNet32, ResNet18, and VGG8 in Section B.1.** We first take ResNet32 as an example. There are three groups of residual blocks in ResNet32. These groups are denoted as “layers” in the implementation, and each “layer” may contain several residual blocks. We interchangeably use “layers” and “groups” in the discussion, and both denote the combination of residual blocks. In ResNet32, residual blocks 1$\sim$5 are encapsulated as group 1; residual blocks 6$\sim$10 are encapsulated as group 2 and 11$\sim$15 as group 3. We treat these groups as the minimal unit when decoupling the network.
>
> In the main paper, we suggest decoupling the network from the last “layer.” We treat the former part as generalized blocks and the latter as specialized ones.  In these experiments, we have explored other kinds of model decoupling, e.g., after the first layer, the second layer, etc. We empirically find that treating the last layer as specialized blocks achieves the best performance with the same memory budget, which is the most memory-efficient. We also conduct experiments to decouple from the middle of the “layer” and find it would not achieve competitive performance as the current strategy. Hence, we recommend decoupling the network at the last layer and treating the last layer as specialized blocks.

---

> ### Author Response · Authors · 2022-11-15
> **Response to Reviewer FkFJ (1/4)**
>
> We thank the reviewer for the feedback and comments. We respond to the concerns below:
>
> **Q1** The CL definition is unclear, making it unclear whether or not the authors employ task-ids during inference. If they use the same definition as Yan et al., 2021. One can assume they do not use the task id. However, it would be helpful to make it clear.
>
> **A1** We thank the reviewer for the suggestion. This paper aims for Class-Incremental Learning (CIL), which does not require the task id during inference. Therefore, **we have emphasized this setting in Section 3.1 in the revised version.**
>
> **Q2** "… Since a single backbone can depict a limited number of features, learning and overwriting new features will undoubtedly trigger the forgetting of old ones …" I would be careful with these kinds of statements. It may seem accurate at first, but we do not know how many concepts can be stored in each feature. For example, the model rarely used the entire weights based on the Lottery Tickets Hypothesis. Therefore, treating this as a limitation without giving further reasons is complex.
>
> **A2** We thank the reviewer for the suggestion. In fact, the “feature overwriting” phenomenon is a specific case of catastrophic forgetting, as discussed in [1]. It should be noted that the Lottery Tickets Hypothesis aims for the scenario where the label space of the full model and pruned model is the same. However, in class-incremental learning, the model faces evolving data, and the supervision signal (label space) changes from time to time. Since the representation ability of a single network is limited, fitting the current model for new tasks will change its characteristics. We can take “finetuning,” a typical baseline method in class-incremental learning [2] which directly finetunes the model with new incremental tasks as an example. When sequentially finetuning with CIFAR100 Base0 Inc10 tasks, the accuracy drastically drops from 85% to 45%. If the Lottery Tickets Hypothesis works for the current situation and the model rarely uses the entire weights, then finetuning can utilize the rest weights for new tasks without harming the former ones. By contrast, it works poorly in the current scenario, and the performance of former tasks drops quickly, which verifies the correctness of our claim.
>
> **Q3** In implementation details, it would be helpful to know how many times the experiments were run. Running the experiments only once is not good practice in CL, since the variance between different initializations and task orders can be significant.
>
> **A3** We thank the reviewer for the suggestion. In the benchmark class-incremental learning setting [2-11], experiments are done by setting the random seed to 1993 to shuffle class orders (as stated in Section 4.1). As a result, we follow the benchmark setting for a fair comparison to former works in the main paper. **Additionally, we have also supplied the evaluations with five runs in Section C.3 in the initial submission,** where we find our proposed method robustly outperforms other methods given different class orders.
>
> Following the suggestions of the reviewer, **in this revision, we have supplied the incremental performance results in Figure 25 (b) of Section C.3.** We show the average and standard mean of each method among five runs, where the conclusions are consistent with the main paper that MEMO robustly works better than other counterparts.
>
> **Q4** The motivation of Fig1 is straightforward but may be out of place. Several things are not explained until Sec4 (for example, Base50, Inc5, or MEMO). The explanation in the text could be improved or Fig1 could be left in Sec4.
>
> **A4** We thank the reviewer for the suggestion. Figure 1 is the core motivation of our research, and putting it on the introduction page can draw the attention of the readers. For example, it shows that exemplar-based and model-based methods have different advantages at different memory scales and triggers the discussion about fairly comparing these methods, a core problem in this paper.
>
> Following the suggestions of the reviewer, **in this revision, we have added illustrations about the terminology like Base50, Inc5, and MEMO in the caption of Figure 1.**

---

### Official Review · Reviewer_dAkF · 2022-10-24

**Confidence:** 4
**Correctness:** 3
**Technical Novelty And Significance:** 4
**Empirical Novelty And Significance:** 4
**Recommendation:** 8

**Clarity, Quality, Novelty And Reproducibility:**

### Clarity & Quality

This paper is well-written and easy to follow.

### Novelty

Evaluating the memory allocation between the exemplars and model parameters is an interesting direction.

### Reproducibility

The authors provide detailed configurations in the appendix.

**Strength And Weaknesses:**

### Strengths

- It is very meaningful to explore how to balance the memory budget usage between exemplars and model parameters in CIL. Saving the model parameters definitely requires a memory budget.

- This paper is well-written and easy to follow.

- Extensive visualizations and figures are provided.

### Weaknesses

- It would be better to discuss some strategies that help to decide which blocks should be frozen. In Figure 5, we can observe that different blocks should be frozen in different settings (e.g., Base0 vs. Base 50). However, we cannot know these settings in advance in real-world applications. Thus, it is important to design some methods that help us to decide how many blocks should be frozen.

- It would be better to include a table in the main paper to show the accuracies of different baselines. Currently, Figure 6 shows the Top-1 accuracy. However, it is more clear to have a table with numerical results.


**Summary Of The Paper:**

In this paper, the authors explore how to balance the memory budget usage between exemplars and model parameters in class-incremental learning. Based on their findings, the authors propose a simple yet effective class-incremental learning method named Memory-efficient Expandable Model (MEMO). MEMO extends specialized layers based on the shared generalized representations, efficiently extracting diverse representations with modest cost and maintaining representative exemplars. Extensive experiments on benchmark datasets validate MEMO’s competitive performance.

**Summary Of The Review:**

Overall, I think this is a high-quality paper. It is well-written and easy to follow. The task studied is very meaningful in my view. I recomannd acceptance.

---

> ### Author Response · Authors · 2022-11-15
> **Thank you for your detailed, positive, and encouraging review**
>
> We thank the reviewer for the detailed, positive, and encouraging review. We are pleased that the reviewer voted our paper for acceptance. We respond to the concerns below:
>
> **Q1** It would be better to discuss some strategies that help to decide which blocks should be frozen. In Figure 5, we can observe that different blocks should be frozen in different settings (e.g., Base0 vs. Base 50). However, we cannot know these settings in advance in real-world applications. Thus, it is important to design some methods that help us to decide how many blocks should be frozen.
>
> **A1** We thank the reviewer for the suggestion. From Figure 5, we can summarize that the specialized layers should be frozen to get better results. Hence, the left problem is whether to freeze or not the generalized layers given the first task training set. **In this revision, we have provided a preliminary experiment on CIFAR100 by varying the number of base classes from \{5,10,20,25,30,35,40,50\} in Section C.2.**
>
> We denote $\bar{\phi(s)}\phi(g)$ as “dynamic generalized blocks,” and $\bar{\phi(s)}\bar{\phi(g)}$ as “frozen generalized blocks” for ease of discussion. Since few base classes are insufficient to obtain a generalizable feature, training with dynamic generalized blocks enables the model to adjust its representation with data evolves. As a result, we observe that the performance of dynamic generalized blocks outperforms frozen generalized blocks at the beginning. However, when increasing the number of base classes, the gap between these strategies becomes smaller, and we observe an intersection when the base class number equals 20. Afterward, freezing generalized blocks shows to be a better strategy for incremental learning, which outperforms dynamic generalized blocks robustly. Hence, in this specific case, we can treat 20 as the empirical threshold to define the strategy for model updating.
>
> Designing proper metrics to measure the generalizability of the shallow layers is interesting future work when it comes to real-world applications. A possible solution is to utilize the hold-out or wild data for evaluation. On the other hand, it would be interesting to explore how to strengthen the generalization ability of shallow layers, e.g., via meta-learning [1], self-supervised learning [2], and deep metric-learning [3].
>
> **Q2** It would be better to include a table in the main paper to show the accuracies of different baselines. Currently, Figure 6 shows the Top-1 accuracy. However, it is not clear to have a table with numerical results.
>
> **A2** We thank the reviewer for the suggestion. **In this revision, we have annotated the final performance of each method in Figure 6.** Due to the page limit, **we have reported the detailed incremental performance of each method in Section A.3.**
>
> [1] Meta-transfer learning for few-shot learning. CVPR19
>
> [2] Supervised contrastive learning. NeurIPS20
>
> [3] Deep metric learning with angular loss. ICCV17

---

> > ### Comment · Reviewer_dAkF · 2022-11-20
> > **Thanks for the feedback.**
> >
> > Thanks for the feedback from the authors. My concerns are addressed.

---

### Official Review · Reviewer_GaiS · 2022-10-24

**Confidence:** 5
**Correctness:** 4
**Technical Novelty And Significance:** 4
**Empirical Novelty And Significance:** 4
**Recommendation:** 8

**Clarity, Quality, Novelty And Reproducibility:**

The contributions are significant and do not exist in prior works. The paper is well-written and easy to follow. The details in this paper are enough to reproduce the experimental evaluations.

**Strength And Weaknesses:**

Strengths:
- New empirical findings shall shed light on the class-incremental learning community, e.g., the experimental evaluations in Figure 3 show that deep and shallow layers have different characteristics in CIL. The results are clear and straightforward, which correspond to the intuitive solutions in model expansion.
- Intuitive method to solve CIL problem. Observing that shallow layers yield higher similarities than deeper layers, the authors argue that expanding deep layers is enough for CIL. The extra memory budget for expanding shallow layers can be exchanged for saving exemplars, which is more memory-efficient. Experimental results verify that the current method can boost performance for free with the same budget.
- Novel performance measures to holistically and fairly compare different CIL methods. I agree with the point that current CIL methods are not compared fairly, especially for the model-based methods with multiple backbones. I think the new performance measure and the way of model evaluation shall inspire the community to design fair comparisons in the future.

Weaknesses:
- The current evaluation protocol depends on some specific and unusual network structures, e.g., ConvNet2, ResNet14, etc. As a result, it would be better for the authors to release the systematical codebase for this research in the final version, which will ease the burden when calculating these new performance measures.

**Summary Of The Paper:**

This paper proposes to tackle the class-incremental learning (CIL) problem from a novel perspective. The current class-incremental learning community is somehow chaotic, where more and more methods tend to design extra model components. These additional components are not counted in the total memory budget, resulting in an unfair comparison between different methods. To this end, this paper unifies the comparison of different methods by introducing the extra dimension for comparison, i.e., memory budget. By varying the memory budget of different methods and designing new performance measures, this paper aims to evaluate incremental models holistically. The results are diverse, where strong models shall fail with limited memory budgets. These results inspire the authors to analyze the importance of each part in CIL and design a simple yet effective method that suits the characteristics of CIL models. Vast experiments on benchmark datasets verify the effectiveness of the proposed method.
I really like the way that the authors study the CIL problem. They performed very systematic and reasonable experiments. The second part of the paper, which looks at the characteristics of different layers, is interesting. Also, the proposed approach (MEMO) just decouples the representations and expands the task-specific layers to strike a trade-off between model expansion and exemplar saving. It is a very interesting way to solve the problem.

**Summary Of The Review:**

Finally, I found the paper clear and easy to read. All the experiments are detailed, as well as the thought process behind them. I think this is a thorough study of a very important problem and can improve many papers that are based on it. As a result, I vote for acceptance.

---

> ### Author Response · Authors · 2022-11-15
> **Thank you for your detailed, positive, and encouraging review**
>
> We thank the reviewer for the detailed, positive, and encouraging review. We are pleased that the reviewer voted our paper for acceptance. **We have attached the source code of MEMO as the supplementary material**, and we will include all the configurations to reproduce the results in the final version.

---

### Official Review · Reviewer_aBa9 · 2022-10-24

**Confidence:** 3
**Correctness:** 3
**Technical Novelty And Significance:** 3
**Empirical Novelty And Significance:** 3
**Recommendation:** 8

**Clarity, Quality, Novelty And Reproducibility:**

Quality: Overall quality is high. Related work seems to cover reasonable background information. Figures are nicely presented and easy to read. Visualization, such as Figure7, helps understand the method.

Clarity: Paper is generally well-written with no obvious grammar mistakes. Figure 6 may need some clarification.  I understand how the data are split, but I don't quite sure about the comparison setting. Are the methods compared under the same memory budget? If yes, how did you decide on the memory budget? If not, what are the hyper-parameters of each approach?

Novelty: Novelty is fine. Layer-wise decompose is not new, however,  the benchmark result and empirical observation may be interesting for the community.



**Strength And Weaknesses:**

Strength:

(1)	Reasonable experimental analysis. The authors conduct extensive experiments to investigate the pros and cons of exampler-based methods and model-based methods given the same memory budget. It will also be interesting to know where the regularization-based methods sit.

(2)	Simple method based on interesting observation. Observation of the effect of different layers is informative. Given the observation, MEMO seems to be a reasonable proposal to save memory for the model-based method.


Weakness:

(1)	This paper focuses on vision benchmarks, which is understandable when comparing performance. However, for observational conclusions such as "shallow layers stay unchanged while deeper layers change more during incremental learning," experiments could cover more settings. For example, have the authors looked at Vision Transformer as the transformer is also popular architecture for vision tasks? Have the authors investigated datasets from another domain, e.g., NLP? Does the observation conclusion still hold in various settings?

Questions:

(1)	MEMO only extend specialized layers to save memory budget for exemplars. One choice here is what layers are considered specialized. The last layer? The last two-layer? The authors provide a study in SectionB.1 using ResNet32. Does the result here generalize to other ResNets and other architectures? What would the authors recommend as the rule of thumb in deciding the specialized layer?


**Summary Of The Paper:**

This paper targets the problem of Class-Incremental-Learning with two major contributions. First, the authors propose to evaluate previous methods with aligned memory budgets to fairly compare exampler-based and model-based approaches. Second, the authors conduct empirical observations on the effect of different layers during model updates. Based on the observation, the authors propose MEMO to create deep layers for new tasks to save memory budget.

**Summary Of The Review:**

Overall, this paper could bring useful insight to the community. (1) It provides a practical evaluation perspective to compare orthogonal direction. (2) Its empirical observation on the gradient norm and shifting range could help understand the training dynamic of Class-Incremental-Learning. (3) The proposed method is a simple and intuitive next step for model-based methods.

---

> ### Author Response · Authors · 2022-11-15
> **Thank you for your detailed, positive, and encouraging review (2/2)**
>
> **Q3** Figure 6 may need some clarification. I understand how the data are split, but I don't quite sure about the comparison setting. Are the methods compared under the same memory budget? If yes, how did you decide on the memory budget? If not, what are the hyper-parameters of each approach?
>
> **A3** We thank the reviewer for the suggestion. **In the revised version, we have added the caption for Figure 6 and detailed configurations in Section A.3.** In Figure 6, all the methods are compared under the same memory budget aligned with DER. Since DER requires storing extra backbones, which consumes the largest budget, we equip other methods with exemplars of the same size. For example, in the CIFAR100 Base0 Inc5 setting, there are 20 incremental stages in total, and DER requires saving 19 extra backbones than iCaRL/WA/Replay. Since the memory budget of a ResNet32 backbone is equal to saving 603 CIFAR images, we equip iCaRL/WA/Replay with (19*603) extra exemplars for a fair comparison. In this revision, we have annotated the corresponding memory budget in the figures, added the illustrations in  Figure 6, and reported the detailed configurations in Section A.3.
>
> [1] An Image is Worth 16x16 Words: Transformers for Image Recognition at Scale. ICLR 2021
>
> [2] Adapting BERT for Continual Learning of a Sequence of Aspect Sentiment Classification Tasks. NAACL 2021
>
> [3] BERT: Pre-training of Deep Bidirectional Transformers for Language Understanding. NAACL 2019

---

> > ### Comment · Reviewer_aBa9 · 2022-11-15
> > **Thank you!**
> >
> > Thanks for the detailed reply, and my concerns have been appropriately addressed.
> >
> > Considering these responses and additional experiments, I would like to increase my score.

---

> > > ### Author Response · Authors · 2022-11-16
> > > **Many thanks!**
> > >
> > > We thank the reviewer very much for the prompt and positive response!

---

> ### Author Response · Authors · 2022-11-15
> **Thank you for your detailed, positive, and encouraging review (1/2)**
>
> We thank the reviewer for the feedback and comments. We respond to the concerns below:
>
> **Q1** This paper focuses on vision benchmarks, which is understandable when comparing performance. However, for observational conclusions such as "shallow layers stay unchanged while deeper layers change more during incremental learning," experiments could cover more settings. For example, have the authors looked at Vision Transformer as the transformer is also popular architecture for vision tasks? Have the authors investigated datasets from another domain, e.g., NLP? Does the observation conclusion still hold in various settings?
>
> **A1** We thank the reviewer for the suggestion. Following the suggestions, we explore the Vision Transformer structure in class-incremental learning tasks and the Bert-based model in incremental NLP classification tasks. **We have included the extra experimental evaluations in Section C.10 in the updated manuscript.** In these supplied experimental evaluations, we summarize that the conclusion of "shallow layers stay unchanged while deeper layers change more during incremental learning” can generalize to different network structures like ViT and Bert and different tasks like NLP aspect sentiment classification.
>
> In detail, we utilize the vanilla ViT structure [1] to replace the backbone in the incremental CIFAR B0 Inc10 tasks and choose ViT with six layers. Following the experiments in the main paper, we trace the shift of weights in each Transformer encoder (including the weights of multi-head attention and MLP) and draw the trends in Figure 31.
>
> Similarly, we follow [2] to use Bert [3] for the ASC dataset (the aspect sentiment classification dataset containing 19 products), which is a benchmark NLP task. We also trace the weight shift of Transformer layers in Bert and draw the trends in Figure 31.
>
> As we can infer from these figures, the conclusions drawn in the main paper that "shallow layers stay unchanged while deeper layers change more during incremental learning” can generalize to different backbones and NLP tasks. Please refer to Section C.10 for detailed figures and illustrations.
>
> **Q2** MEMO only extend specialized layers to save memory budget for exemplars. One choice here is what layers are considered specialized. The last layer? The last two-layer? The authors provide a study in SectionB.1 using ResNet32. Does the result here generalize to other ResNets and other architectures? What would the authors recommend as the rule of thumb in deciding the specialized layer?
>
> **A2** We thank the reviewer for the suggestion. We first take ResNet32 as an example. There are three groups of residual blocks in ResNet32. These groups are denoted as “layers” in the implementation, and each “layer” may contain several residual blocks. We interchangeably use “layers” and “groups” in the discussion, and both denote the combination of residual blocks. In ResNet32, residual blocks 1$\sim$5 are encapsulated as group 1; residual blocks 6$\sim$10 are encapsulated as group 2 and 11$\sim$15 as group 3. We treat these groups as the minimal unit when decoupling the network.
>
> In the main paper, we suggest decoupling the network from the last “layer.” We treat the former part as generalized blocks and the latter part as specialized ones. **In this revision, we have supplied experiments about network decoupling on various network structures, including ResNet32, ResNet18, and VGG8 in Section B.1.** In these experiments, we have explored other kinds of model decoupling, e.g., from the first layer, second layer, etc. We empirically find that treating the last layer as specialized blocks achieves the best performance with the same memory budget, which is the most memory-efficient. We also conduct experiments to decouple from the middle of the “layer” and find it would not achieve competitive performance as the current strategy. Hence, we recommend decoupling the network at the last layer and treating the last layer as specialized blocks.

---

### Author Response · Authors · 2022-11-15
**General Response**

We would like to express our deepest gratitude to the reviewers for the meticulous examination of the paper and their insightful and valuable comments. We acknowledge that all the reviewers observed the shining point, saying the suggested protocol for class-incremental learning is **fair, interesting, systematic, and reasonable** (aBa9, GaiS, dAkF, FkFJ). They also consider our proposed method as well as the new performance measures meaningful (dAkf), shedding light on the class-incremental learning community (GaiS), and bringing useful insight to the community (aBa9). Additionally, all the reviewers acknowledge that extensive experiments validate the memory efficiency of our proposed method (aBa9, GaiS, dAkF, FkFJ).

In this rebuttal, we have given careful thought to the reviewers’ suggestions and made the following revisions to our manuscript to answer the questions and concerns:
- In **Sections 1, 2, 3, 4, and 5**, we add illustrations about problem setting, discussions about related work, suggestions about model decoupling, and details about experiment evaluations;

- In **Supplementary Section A.3**, we add numerical results about the benchmark comparison in Figure 6 and corresponding implementation details;

- In **Supplementary Section B.1**, we add the experiments about network decoupling and give the rule of thumb to define specialized and generalized blocks;

- In **Supplementary Section C.2**, we add the experiments about network freezing and discussions about how to enhance the generalizability of shallow layers;

- In **Supplementary Section C.3**, we add the experiments to trace the accuracy of base and new classes and give the intuition about how dynamic generalized blocks help model learning new tasks;

- In **Supplementary Section C.4**, we add the incremental performance of multiple trials, and show MEMO robustly outperforms the state-of-the-art by a substantial margin;

- In **Supplementary Section C.10**, we add the experiments to evaluate class-incremental learning models with ViT and Bert and show the conclusion that “shallow layers change less while deep layers change more” still holds for different networks and datasets;

- We have uploaded the source code of MEMO as the **Supplementary Material**.

We have highlighted the revised part in our manuscript in **blue** color. Please check the answers to specific comments.

---

### Decision · Program_Chairs · 2023-01-20

**Decision:**

Accept: notable-top-25%

**Justification For Why Not Higher Score:**

I think this paper could make a decent oral presentation, however it is fairly narrow in scope as it looks at a specific problem in class-incremental learning and makes no general, technical contribution (the methodological contributions are its strength). Thus I feel it has somewhat limited appeal to the broader ICLR audience.

**Justification For Why Not Lower Score:**

The research community in continual learning is coming out of an expansive, breadth-first phase of exploration and experimental search for effective techniques. I think this paper represents an important step towards consolidating results from this exploration phase, a step that should provide a more solid foundation for moving forward.

**Metareview: Summary, Strengths And Weaknesses:**

# Summary of Contribution

This paper addresses the trade-off between memory budget and performance in class-incremental learning (CIL). The authors begin with a critical examination of how overall CIL memory budget should be determined by considering both model parameters and exemplar memory size. They propose a simple (but effective) baseline for their comparisons based on an expandable memory (MEMO). They also propose new metrics to analyze the trade-off between model size and exemplar memory in terms of CIL performance. Extensive and critical experimental results are given for MEMO and a variety of consolidated baselines from the state-of-the-art. The experimental comparison is given on small- (CIFAR) and large-scale image (ImageNet) recognition benchmarks.

# Strengths

+ **Clarity**: The paper is very well-written and the main technical points and arguments are well-motivated from first principles. The visualizations are well thought-out and complement the analyses and experimental results throughout.

+ **Experimental Analysis**: The main contribution of the work is in the empirical analysis of the model/exemplar memory trade-off. The experiments and accompanying analyses shed light on one of the key issues in class-incremental learning, and although the problem of memory/model trade-off is well-known this seems to be the first very thorough and decisive exploration of this aspect.

+ **Impact**: The metrics and new baselines proposed in the paper are likely to have lasting effect on the broader community involved in incremental learning research (and potentially also in practice). Though not entirely comprehensive in its approach, this work represents an excellent starting point for deeper understanding of the many trade-offs in continual learning.

# Weaknesses

+ **Non-vision Benchmarks and Other Architectures**: The paper exclusively addresses vision benchmarks using CNN backbones. During the discussion period the authors performed additional experiments using Transformers (both ViT and BERT), providing some evidence that the main conclusions of the paper hold also for other tasks and backbone architectures.

+ **Mitigation Specifics**: The contribution would be stronger if the experimental analyses delved deeper into more specifics on *how* to divide between specific and general layers.

+ **Other Orthogonal Directions**: This paper looks at one specific trade-off in class-incremental learning. A discussion on how other types approaches (e.g. regularization) could be brought into the same analytical framework.

# Summary

The reviewers are nearly unanimous in their very positive evaluation of the paper. During a very lively back-and-forth during the discussion period the authors addresses all of the main reviewer concerns, provided selected new experiments on new tasks and using new backbones, and in general significantly improved the presentation and clarity of the underlying contributions of the work. The research community in continual learning is coming out of an expansive, breadth-first phase of exploration and experimental search for effective techniques. This work attempts to abstract, generalize, and accurately measure one specific aspect of these recent advances, which is something very useful to the entire community working on class-incremental learning.


**Note From Pc:**

if the above contains the word "oral" or "spotlight" please see: "oral" presentation means -> notable-top-5% and "spotlight" means -> notable-top-25%. As stated in our emails, we are disassociating presentation type from AC recommendations